# TOPO-FIELD: TOPOMETRIC MAPPING WITH BRAIN-INSPIRED HIERARCHICAL LAYOUT-OBJECT-POSITION FIELDS

## ABSTRACT

Mobile robots require comprehensive scene understanding to operate effectively in diverse environments, enriched with contextual information such as layouts, objects, and their relationships. While advancements like Neural Radiance Fields (NeRF) offer high-fidelity 3D reconstructions, they are computationally intensive and often lack efficient representations of traversable spaces essential for planning and navigation. In contrast, topological maps generated by LiDAR or visual SLAM methods are computationally efficient but lack the semantic richness necessary for a more complete understanding of the environment. Inspired by neuroscientific studies on spatial cognition, particularly the role of postrhinal cortex (POR) neurons that are strongly tuned to spatial layouts over scene content, this work introduces Topo-Field, a framework that integrates Layout-Object-Position (LOP) associations into a neural field and constructs a topometric map from this learned representation. LOP associations are modeled by explicitly encoding object and layout information, while a Large Foundation Model (LFM) technique allows for efficient training without extensive annotations. The topometric map is then constructed by querying the learned implicit neural representation, offering both semantic richness and computational efficiency. Empirical evaluations in multi-room apartment environments demonstrate the effectiveness of Topo-Field in tasks such as position attribute inference, query localization, and topometric planning, successfully bridging the gap between high-fidelity scene understanding and efficient robotic navigation.

## 1 INTRODUCTION

Mobile robots are rapidly moving from research labs to widespread use. For these robots to operate autonomously in complex environments, a deep understanding of their surroundings is crucial (Cadena et al., 2016). Efficient path planning and accurate identification of navigable spaces, along with detailed environmental reconstruction, will be key to enabling their deployment in real-world scenarios (Blochliger et al., 2018).

Recently, detailed environmental reconstruction has made great progress in producing lifelike 3D reconstructions (Ullman, 1979; Forster et al., 2014; Dai et al., 2017; Tang & Tan, 2018), in which NeRF (Mildenhall et al., 2020) is a prime instance. As improvements, works like (Zhi et al., 2021; Fan et al., 2022; Xie et al., 2021) introduce semantic information for better scene understanding. Further, features powered by Large-Foundation-Model (LFM)s, trained on massive datasets across various scenes, are employed with general knowledge for open scene understanding (Shafiullah et al., 2022; Huang et al., 2023; Kerr et al., 2023). However, it is computationally demanding and lacks global layout information using detailed neural fields for planning and navigation.

In contrast, existing topological maps for path planning and navigation in complex environments are often derived from LiDAR Simultaneous-Localization-and-Mapping (SLAM) using 3D dense submaps (Gomez et al., 2020) or visual SLAM by clustering free-space regions and extracting occupancy information from point clouds (Blochliger et al., 2018). While this approach increases path planning accuracy, computing topology with traditional methods comes with high computational costs and tends to strip away essential semantic information, reducing the robot's ability to fully

Figure 1: **Illustration of the Topo-Field strategy and capabilities.** Hierarchically dividing scene information into layout, object, and position to model them explicitly, layout-object-position associated knowledge enables robots with a topometric map representing the scene and planning navigable path to realize a more comprehensive spatial cognition.

understand and interpret the environment, which is critical for advanced autonomous functions such as language/image-prompted localization and navigation.

Neuroscientists have long discovered that animals process their surroundings using topological coding, forming what is known as a "cognitive map" (Tolman, 1948), a concept embodied by place cells (O'Keefe & Dostrovsky, 1971). These place cells, along with spatial view cells (Rolls et al., 1998), respond to specific scene contents. More recently, research has shown that a population code in the postrhinal cortex (POR) is strongly tuned to spatial layout rather than scene content (LaChance et al., 2019), capturing spatial representations relative to environmental centers to form a high-level cognitive map from egocentric perception to allocentric understanding (Zeng et al., 2022), unlike traditional clustering from occupancy information (Blochliger et al., 2018) or Voronoi diagrams (Friedman et al., 2007). Inspired by this, we intuitively abstract the neural representations of space to build topo-field in three key aspects: 1) The cognitive map corresponds to a topometric map, which uses graph-like representations to encode relationships among its components, e.g. layouts and objects. 2) The population of place cells is analogous to a neural implicit representation with position encoding, enabling location-specific responses. 3) POR, which prioritizes spatial layouts over content, aligns with our spatial layout encoding of connected regions. We believe this approach makes a step forward in applying mechanisms of spatial cognition in robotics.

To this end, this work proposes a Topo-Field, integrating the Layout-Object-Position (LOP) association into neural field training and constructing a topometric map based on the learned neural implicit representation for hierarchical robotic scene understanding. By inputting RGB-D sequences, objects and background contexts are encoded separately as contents and layout information to train a neural field, forming a detailed scene representation. A contrast loss against features from LFMs is employed, resulting in little need for annotation. Further, a topometric map is built based on co-observation relationship among frames, sampling points, and querying the learned field, which is efficient for navigable path planning. To validate the effectiveness of Topo-Field, we conduct quantitative and qualitative experiments on several multi-room apartment scenes evaluating the abilities including position attributes inference, text/image query localization, and planning.

Our contributions can be listed as follows:

- We develop a brain-inspired Topo-Field, which combines detailed neural scene representation with high-level efficient topometric mapping for hierarchical robotic scene understanding and navigable path planning. Various quantitative and qualitative experiments on real-world datasets are conducted, showing high accuracy and low error in position at-

tributes inference and multi-modal localization tasks. Examples of topometric construction and path planning are also employed.

- We explain the theoretical basis and neuroscience reference to manage the hierarchical encoding of spatial layouts and contents in the form of objects and connected regions, according to the spatial mechanism of cognitive map with POR population and place cells.

- We propose to learn a Layout-Object-Position associated implicit neural representation with target features from separately encoded object instances and background contexts as objects and layouts. The process is explicitly supervised by LFM-powered strategy with little human labor.

- We propose a topometric map construction pipeline by querying the learned neural representation in a two-stage mapping and updating approach, leveraging LLM to validate edges conducted among vertices.

## 2 RELATED WORKS

### 2.1 DENSE REPRESENTATION WITH NEURAL RADIANCE FIELD

Detailed 3D scene reconstruction has made great efforts in producing lifelike results, among which NeRF (Neural Radiance Fields) (Mildenhall et al., 2020) has widely attracted researchers' attention. While numerous efforts improve the NeRF (Yu et al., 2021; Martin-Brualla et al., 2021; Zhu et al., 2022), a popular research direction is to integrate semantics with NeRF to achieve a more comprehensive understanding of scenes (Zhi et al., 2021; Fan et al., 2022; Xie et al., 2021). Recently, several robotic works have demonstrated that features from LFMs can be used for self-supervised learning, which reduces the costly manual annotation (Shafiullah et al., 2022; Huang et al., 2023; Kerr et al., 2023). However, the semantic feature fields learned in the above methods focus on object semantics but do not include layout-level features. RegionPLC (Yang et al., 2023) considered region information by fusing multi-model features but with no explicit representation of layout features. In contrast, in our work, CLIP (Radford et al., 2021) and Sentence-BERT (Reimers & Gurevych, 2019) are employed to generate vision-language and semantic features for objects and layout respectively. In addition to using object semantics, we annotate the belonging regions based on spatial layout and regional division of scenes. Such annotations incur minimal cost but establish connections between the position of 3D points, object semantics, and scene regions.

### 2.2 TOPOMETRIC MAP FOR SCENE STRUCTURE UNDERSTANDING

Using detailed neural fields for planning and navigation is computationally demanding, on the other hand, hybrid topometric mapping has been known for its efficiency in terms of managing the information and being queried for downstream tasks (Zhang, 2015; Zhang et al., 2015; Garrote et al., 2018). It takes advantage of both metric maps and topological maps. Metric maps could refine the local scale geometry accuracy and navigation plans while topological maps provide reliable global topological cues and large-scale plans (Oleynikova et al., 2018; Badino et al., 2012). However, most topological maps have not introduced information such as semantics. This makes it unsuitable for language/image-guided planning tasks, which is a growing trend in scene representation applications. Concept-graph (Gu et al., 2024) makes a step forward utilizing LFM to model the object structure with a topo map. CLIO Maggio et al. (2024)built a task-driven scene graph inspired by Information Bottleneck (IB) principle to form task-relevant clusters of primitives. At the same time, HOV-SG Werby et al. (2024) proposed a hierarchical scene understanding pipeline, using feature point cloud clustering of zero-shot embeddings in a fusion scheme and realizing the mapping in an incremental approach. Unlike the incremental mapping and clustering-based graph construction method, we propose to build the topometric map based on querying the trained neural field which serves as knowledge-like memory base, whose nodes and edges include attributes representing object and layout information explicitly learned when training the specific neural representation encoding.

## 2.3 Spatial Understanding with Layout Information

Generally, topology is built based on traditional clustering from occupancy information or Voronoi diagrams (He et al., 2021), regardless of the contents and layout relationship. However, neuroscience findings suggest a mechanism to form a high-level cognitive map from egocentric perception to allocentric representation (Zeng et al., 2022). Neuroscientists have long discovered that animals process their surroundings using topological coding, forming a "cognitive map" (Tolman, 1948). Place cells (O'Keefe & Dostrovsky, 1971), as the embodiment, together with spatial view cells show activity to contents (Rolls et al., 1998). Recently, Patrick et al. (LaChance et al., 2019) showed that a population code in the POR is more strongly tuned to the spatial layout than to the content in a scene. This suggests that there are specialized cells and signaling mechanisms to process layout in the process of scene understanding, which captures the spatial layout of complex environments to rapidly form a high-level cognitive map representation (Zeng et al., 2022). Inspired by the above research, we propose that the spatial layout connected by regions, as a high-level abstract feature, is closely related to the object contents and purposes of the scene. We mimic the neural scene understanding mechanism by employing egocentric neural field knowledge to construct allocentric topometric map.

## 3 Overview

We propose to learn an implicit representation of a scene with the neural encoding approach by establishing associations between 3D positions and their corresponding layout and object features as the scene knowledge. Then, a topometric map is built with the learned neural field to form an efficient and queriable representation with a comprehensive understanding of the scene. Therefore, we need to train a scene-dependent implicit function, denoted as

$$F : \mathbb{R}^3 \to \mathbb{R}^n, \tag{1}$$

where for any 3D point $P$ in space, $F(P)$ is supposed to match with

$$\mathcal{E}\{(e_v, e_s)\} \in \mathbb{R}^n, \tag{2}$$

representing the layout-object-position associated embedding of that point. $e_v$ and $e_s$ are vision-language embedding and semantic embedding of image point where $P$ is back-projected from. CLIP (Radford et al., 2021) image encoder is introduced to encode $e_v$ integrating the vision and language feature space. Besides, the Sentence-BERT (Reimers & Gurevych, 2019) feature is also introduced to encode $e_s$ in this work. Because intuitively, unlike objects that can have similar appearances within a certain category, region information often lacks specific visual appearances and is closely related to semantic representations like the integration purpose of the scene and object semantics. Models trained on large-scale question-answering datasets can aid in understanding the semantic relationships between regions and objects. Target feature processing and training strategy to match the embeddings to targets are described in Section 4.1 and 4.4. Applications utilizing the learned field are discussed in Section 4.3.1. Based on the trained $F$, we aim to build a topometric map denoted as

$$G = (V, E), \tag{3}$$

where vertices $V$ include object vertices $\mathbf{v}_o$ and region vertices $\mathbf{v}_r$ and edges $E$ include edges between objects $\mathbf{e}_{o-o}$, edges between regions $\mathbf{e}_{r-r}$, and edges between object and region $\mathbf{e}_{o-r}$. The topological map architecture and construction pipeline are described in Section 4.3.2.

## 4 Method

### 4.1 Target Feature Processing

(We clarify the formulation and polish for better understanding. The ground-truth label of layout regions is described. More details are described for pixel-wise encoding of image and information in the target supervising embeddings.)

RGB-D image sequences with poses are accepted as input to get the target layout-object-position features for training $F$. For pure RGB image sequences, depth point clouds and camera poses can also be estimated through methods like COLMAP (Schönberger & Frahm, 2016) or simultaneous

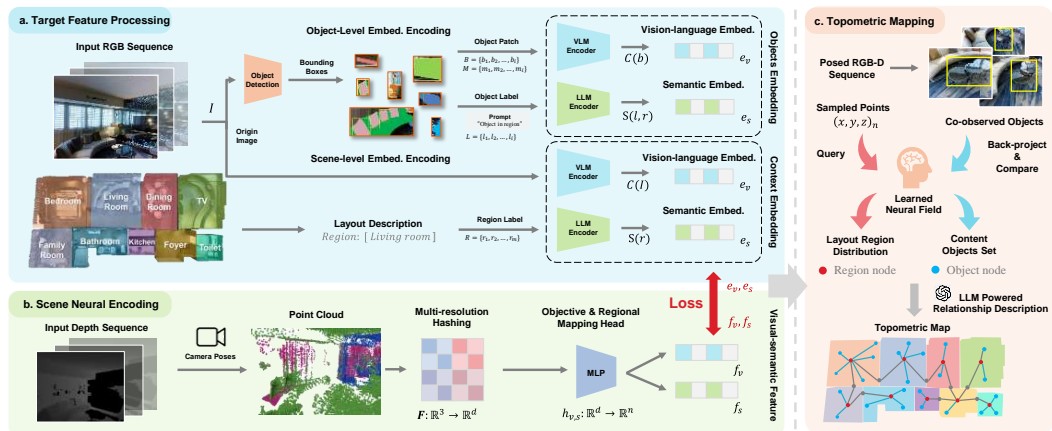

Figure 2: **Pipeline of the Topo-Field.** **(a)** The ground truth generation of layout-object-position vision-language and semantic embeddings for weakly-supervising. **(b)** The neural implicit network mapping 3D positions to target feature space. A contrastive loss is optimized against each other. **(c)** Topometric mapping process with trained neural field. (Formulation in the figure has been clarified.)

localization and mapping (SLAM). The only employed GT annotation is the layout distribution of environment where the region of each 3D point $P$ is denoted as $r_P \in R = \{r_1, r_2, \ldots, r_q\}$, where $q$ is the number of regions. Such information is available in datasets like Matterport3D Chang et al. (2017). However, in fact, partitioning the buildings needs little human labor, where in most human-made buildings spatial layouts are easily available divided by straight walls if not provided. As in our practice, region annotation of a house with 8 rooms only takes 3 min by drawing lines from top-down view according to walls to form a rule to separate $(x, y)$ coordinates, bounding 3D points to different regions.

For each image $I$, we employ Detic (Zhou et al., 2022) $D$ to generate object instance patches with number $i$, including bounding-boxes $B = \{b_1, b_2, \ldots, b_i\}$, masks $M = \{m_1, m_2, \ldots, m_i\}$, and labels $L = \{l_1, l_2, \ldots, l_i\}$.

For object pixels $p_o$ in instance mask $j$, CLIP (Radford et al., 2021) $C$ is employed to compute per-pixel features in mask $b_j$ and Sentence-BERT (Reimers & Gurevych, 2019) $S$ is employed to process the semantic feature of $l_j$, prompted in the form of "$l_j$ in $r_{p_o}$". Given the related region $r_{p_o}$ of $p_o$, embedding of $p_o$ can be denoted as $e_{p_o} = \{C(b_j), S(l_j, r_{p_o})\}$.

What's more, the background appearance is also considered which we proposed to include context information for region layout. For background pixels $p_b$ out of masks, per-pixel feature of the whole image $I$ is encoded. Its related region $r_{p_b} \in R = \{r_1, r_2, \ldots, r_m\}$ is regarded as the text label and embedding of $p_b$ can be calculated as $e_{p_b} = \{C(I), S(r_{p_b})\}$.

Then, pixel-wise embeddings are back-projected to 3D space based on depth and pose and averagely counted to form a distilled 3D feature point cloud. Consequently, the target feature space $\mathcal{E}\{(e_v, e_s)\}$ consists of object and layout features, where $(e_v, e_s)$ directs from $\{e_{p_o}, e_{p_b}\}_{p_o, p_b \in P}$. The pipeline is shown in Fig. 2

Compared with previous implicit neural field methods, $(e_v, e_s)$ includes (1) separately encoded vision-language and semantic information by supervising embeddings from object and background pixels. (2) region information consisted of vision-language embeddings from per-pixel image encoding and semantic embeddings from region text labels. (3) context included object label in the form of "$l_p$ in $r_p$", where $l_p$ and $r_p$ is object label and region label at point $p$ (e.g., cup in the kitchen). Ablation studies of these improvements are conducted in Section 4 with more details.

## 4.2 SCENE NEURAL ENCODING

(We clarify the formulation and include more details of the MHE and feature mapping head to get the high dimension feature in neural representation.)

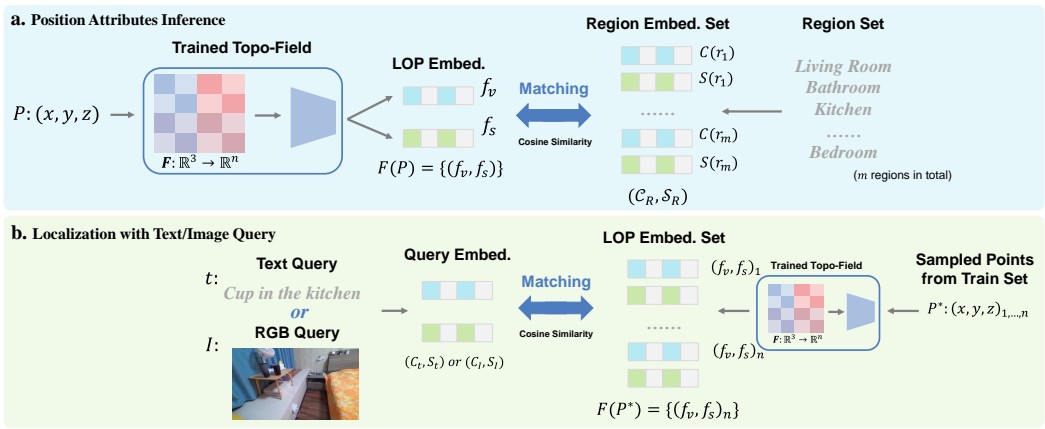

Figure 3: **Capabilities of the learned neural field**. **(a)** The attributes inference using position input. **(b)** The LOP association helped localization of text and image queries. (Formulation in the figure has been clarified.)

Our proposed Topo-Field involves an implicit mapping function to encode the 3D position into a spatial vector representation $g : \mathbb{R}^3 \to \mathbb{R}^d$ and separate heads $h : \mathbb{R}^d \to \mathbb{R}^n$ processing encodings to match the target feature space $\mathcal{E}\{(e_v, e_s)\}$. To select an appropriate implicit function, considering that the target feature space includes object-level local features and layout-level region feature representation, we employ the Multi-scale Hash Encoding (MHE) introduced in Instant-NGP (Müller et al., 2022) as $g$ with $d = 144$. The feature pyramid structure used in MHE allows for considering structural features ranging from coarse to fine in the spatial domain. Additionally, MHE has a faster training speed compared to traditional NeRF (Mildenhall et al., 2020) network structures. For mapping the position encodings to the target feature space, we employ a unified and simple Multi-Layer Perceptron (MLP) network structure. It includes heads $h_v : \mathbb{R}^d \to f_v$ for obtaining vision-language features and $h_s : \mathbb{R}^d \to f_s$ for semantic features, which together form the high dimension embeddings $\{f_v, f_s\} \in \mathbb{R}^n$. The model is shown in Fig. 2.

In this way, given a posed RGB-D image, the target feature of each pixel is processed as mentioned in Section 4.1 denoted as $\mathcal{E}\{(e_v, e_s)\}$. At the same time the related pixel in depth image is backprojected into 3D space according to depth and pose value and processed by the above mentioned $g, h$ to form $\{f_v, f_s\}$. A contrastive loss is conducted between $\{(e_v, e_s)\}$ and $\{f_v, f_s\}$ to train the neural representation. Training details are declared in Section 4.4.

## 4.3 TOPOMETRIC MAPPING

With the function and feature representation mentioned above, we can integrate 3D positions with the object and region information and construct a topometric map. The topo map construction process is formed in a mapping and updating strategy, while the implicit neural representation is introduced and queried as scene knowledge in this process. Detailed pipeline is introduced as follows.

### 4.3.1 KNOWLEDGE FROM LEARNED NEURAL FIELD

(We clarify the formulation and definition for better understanding.)

**Position Attributes Inference.** Using spatial 3D point $P$ as input, assuming a collection of space regions $R$ (e.g., "living room""bathroom""bedroom"...), we compute the vision-language features $\mathcal{C}_R = \{C(r_1), C(r_2), \ldots, C(r_m)\}$ and semantic features $\mathcal{S}_R = \{S(r_1), S(r_2), \ldots, S(r_m)\}$ using CLIP Radford et al. (2021) encoder $C$ and Sentence-BERT Reimers & Gurevych (2019) encoder $S$, where $m$ is the number of rooms. Then the cosine similarity between $F(P) = \{(f_v, f_s)\}_P$ and $\{\mathcal{C}_R, \mathcal{S}_R\}$ is calculated to find the most likely region to which $P$ belongs. The inference process is shown in Fig. 3 (a). Similarly, the object information of $P$ can be inferred with the same approach replacing the region set $R$ with object set $O$.

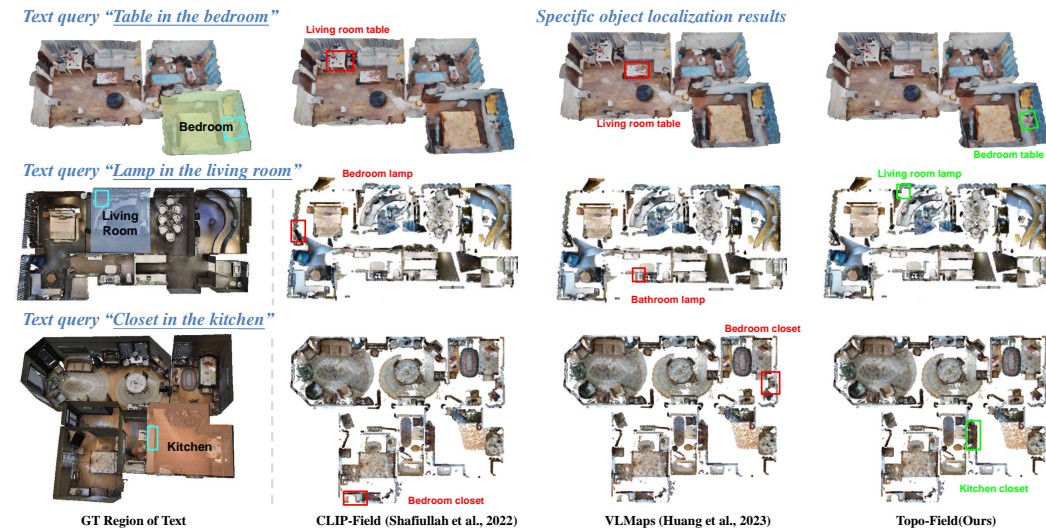

Figure 4: **Qualitative comparison of text query localization** results among state-of-the-art methods and our method with text input in the form of "*object in the region*". Blue box shows the ground truth bounding box of object. Red box means miss-predicted box, while green box means the correctly predicted results.

| Methods | Scene1 | | Scene2 | | Scene3 | | Scene4 | |
|---|---|---|---|---|---|---|---|---|
| | Dist. | Acc. | Dist. | Acc. | Dist. | Acc. | Dist. | Acc. |
| CLIP-Field(2022) | 2.97 | 0.24 | 3.35 | 0.21 | 2.98 | 0.20 | 3.06 | 0.17 |
| VLMaps(2023) | 2.78 | 0.28 | 3.63 | 0.16 | 3.05 | 0.24 | 3.12 | 0.12 |
| LERF(2023) | 2.86 | 0.32 | 2.82 | 0.11 | 3.49 | 0.17 | 3.04 | 0.20 |
| Topo-Field | **0.92** | **0.85** | **0.86** | **0.84** | **0.36** | **0.95** | **0.27** | **0.97** |
| Text queries | 100 | | 100 | | 60 | | 60 | |

Table 1: **Quantitative comparison of text query localization** results on different scenes from the Matterport3D dataset. The average distance (m) from the target to the localized point cloud and the accuracy evaluating whether predicted positions are in the correct region are used as metrics.

**Localization with Text/Image Query.** For natural language text input $t$ (e.g., "cup in the bedroom"), most existing robotic scene representations struggle to locate specific objects of interest (e.g., differentiating between cups in the living room and the bedroom). However, with our proposed Topo-Field that includes region information, we can calculate the cosine similarity between $\{\mathcal{C}_t, \mathcal{S}_t\}$ and the embeddings $F(P^*) = \{(f_v, f_s)\}_{P^*}$ to find the most likely position of queries, where $P^*$ are sampled from 3D points set to train $F$. As for image input $I$, we can calculate the cosine similarity of $\{\mathcal{C}_I, \mathcal{S}_I\}$ with $F(P^*) = \{(f_v, f_s)\}_{P^*}$ in the same way to find the 3D points set with highest similarity. The localization process of both text query and image query is shown in Fig. 3.

### 4.3.2 TOPOMETRIC MAP CONSTRUCTION

(We describe the topometric map construction pipeline step-by-step with more details, which can be regarded as a mapping and updating strategy. Definition and attributes of vertices and edges are clarified, including the acquisition process.)

As defined in Section 3, topometric map $G = (V, E)$ consists of vertices and edges. We define a vertice **v** and edge **e** as

$$\mathbf{v} : \{ \text{id, node\_type, class, bounding\_box, caption}\}, \tag{4}$$

$$\mathbf{e} : \{ \text{id, edge\_type, start\_node, end\_node, relationship, caption }\}. \tag{5}$$

Mimicking the mental representation of cognitive maps, we construct the topometric map in a **mapping and updating** strategy based on the learned Topo-Field $F$.

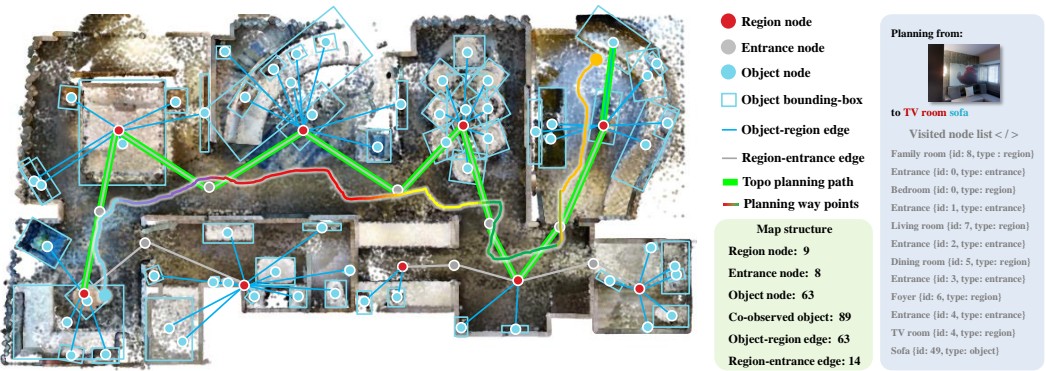

Figure 5: **Topometric map construction example.** The topometric map is represented as a graph from a top-down view according to the position of nodes. Map structure shows number of nodes and edges. A planning path from a seen view to target is shown as an example employing topometric map, the path is highlighted in green showing the related nodes and edges. Visited nodes are listed on the right. The line with gradient colors represents the waypoints based on the planning results while different colors represent different predicted regions of waypoints.

**Mapping**. we first averagely sample $k$ points $P_{1,\dots,k}$ in the environment (each grid of $0.5m \times 0.5m$ with a point in our practice) and infer their related regions according to Section 4.3.1. Supposing there are $m$ regions in total $r_{1,\dots,m}$, we calculate the extent of each region in the bounding-box format according to positions of points within the same region. The topo map region vertice set is then initialized as $\mathbf{v}_r = \{\mathbf{v}_{r_1}, \mathbf{v}_{r_2}, \dots, \mathbf{v}_{r_m}\}$. For each $\mathbf{v}$, {id} is set, {node_type} is {region}, {class} and {caption} is set according to the inferred region label, and {bounding_box} is set to

$$\{[Min(x), Max(x)], [Min(y), Max(y)], [Min(z), Max(z)]\}, (x, y, z) \in P_{1,\dots,k}. \quad (6)$$

On the other hand, while employing Detic Zhou et al. (2022) to detect object instances as mentioned in Section 4.1, instances with high confidence (more than 60% in our practice) are recorded as object vertices candidates. For each $\mathbf{v}$, {node_type} is {object}, {class} and {caption} is set according to the prediction result, and {bounding_box} is set according to the back-projected masked pixels similar to equation 6. With the mapped nodes, we leverage LLM to describe the layouts with connectivity, distances, and relationships of regions and objects in JSON format based on the vertices' attributes and poses. During this process, edges are built among vertices. For object-object edge $\mathbf{e}_{o-o}$, we follow Gu et al. (2024) which mainly consider bounding-box overlap. For object-region edge $\mathbf{e}_{o-r}$, we consider an object belongs to the region if the object b-box is in the region b-box and filter the unreasonable relation noise powered by LLM (e.g., it's almost impossible that a bike is in bedroom). For region relationships, the adjacency and position relationship of region b-box is considered. Examples of LLM prompts to build relationships and JSONs are listed in appendix for reference. Fig. 2 shows the pipeline of metric-topological map construction.

**Updating**. RGB-D image sequence for training $F$ or a newly captured sequence can be used for constructed topometric map fine-tuning. For object vertices, if an object is detected by more than 3 frames in sequence, the object b-box will be compared with the constructed vertices. A new vertice will be added if no vertice corresponds to it with the above-mentioned process. For region vertices, we calculate embeddings $F(p_I)$ of sampled back-projected pixels $p_I$ in each image $I$. $F(p_I)$ will be matched with the constructed region set $r_{1,\dots,m}$, and extent of a region $r$ will be updated if $F(p_I)$ matches $\{\mathcal{C}_r, \mathcal{S}_r\}$ and $p_I$ exceeds the {bounding_box} extent of vertice $\mathbf{v}_r$. LLM to update edges will be called each 50 frames.

## 4.4 TRAINING

The pipeline of ground truth data generation is described in Section 4.1 to train $F$. To fit the implicit representation introduced in Section 4.2 to the target feature space, we design the loss function through a contrastive approach. For the vision-language feature optimization, the tempered similarity matrix on point $P$ is denoted as

$$\text{Sim}_v = \tau\{f_v\}_P\{e_v\}_P, \quad (7)$$

| Methods | Scene1 | Scene2 | Scene3 | Scene4 | Scene5 | Scene6 | Scene7 | Scene8 | Scene9 | Scene10 |
|---|---|---|---|---|---|---|---|---|---|---|
| CLIP-Field(2022) | 0.242 | 0.165 | 0.130 | 0.142 | 0.127 | 0.138 | 0.227 | 0.200 | 0.102 | 0.060 |
| VLMaps(2023) | 0.177 | 0.194 | 0.127 | 0.098 | 0.148 | 0.187 | 0.199 | 0.221 | 0.092 | 0.087 |
| LERF(2023) | 0.268 | 0.189 | 0.165 | 0.153 | 0.136 | 0.169 | 0.216 | 0.252 | 0.110 | 0.091 |
| RegionPLC(2023) | 0.290 | 0.202 | 0.173 | 0.168 | 0.152 | 0.154 | 0.243 | 0.248 | 0.086 | 0.088 |
| Topo-Field | **0.886** | **0.900** | **0.884** | **0.894** | **0.872** | **0.858** | **0.901** | **0.897** | **0.821** | **0.839** |
| Position Samples | 169k | 185k | 111k | 112k | 106k | 176k | 130k | 121k | 205k | 211k |

Table 2: **Comparison of position attributes inference results** on the test set of different scenes from the Matterport3D dataset. The average region prediction accuracy of sampled 3D points is used as metric.

| Methods | Scene1 | Scene2 | Scene3 | Scene4 |
|---|---|---|---|---|
| CLIP-Field(2022) | 2.541 | 2.748 | 2.922 | 2.651 |
| VLMaps*(2023) | 2.112 | 1.894 | 1.181 | 1.595 |
| LERF(2023) | 1.276 | 1.175 | 1.148 | 1.129 |
| Topo-Field | **0.742** | **0.830** | **0.374** | **0.327** |

Table 3: **Quantitative comparison of image query localization** results with other methods. The similarity weighted average distance (m) between the target view point cloud and the predicted point cloud is evaluated. VLMaps* is a self-implemented version with image localization ability.

where $\tau$ is the temperature term, $\{f_v\}_P$ and $\{e_v\}_P$ is the calculated implicit representation feature and target embedding according to $P$. Using cross-entropy loss, the vision-language loss can be calculated as

$$\mathcal{L}_v = -exp(-\text{dist}_P)(H(\text{Sim}_v) + H(\text{Sim}_v{}^T)), \tag{8}$$

where $\text{dist}_P$ is the distance from $P$ to camera, and $H$ is the cross-entropy function. For the semantic loss, similarity on points $P$ can be calculated as

$$\text{Sim}_s = \tau\{f_s\}_P\{e_s\}_P. \tag{9}$$

Similarly, semantic loss can be denoted as

$$\mathcal{L}_s = -\text{conf}(H(\text{Sim}_s) + H(\text{Sim}_s{}^T)), \tag{10}$$

where $conf$ is the prediction confidence from the detection model. The total loss is computed by:

$$\mathcal{L} = \mathcal{L}_v + \mathcal{L}_s. \tag{11}$$

In our experiments, an NVIDIA RTX3090 GPU is utilized and the batch size is set to $12544$ to maximize the capability of our VRAM. As model instances, CLIP with SwinB is employed in Detic Zhou et al. (2022), CLIP Radford et al. (2021) encoder is ViT-B/32 and Sentence-BERT Reimers & Gurevych (2019) encoder is all-mpnet-base-v2. The MHE has 18 levels of grids and the dimension of each grid is 8, with $log_2$ hash map size of 20 and only 1 hidden MLP layer of size 600. We train the neural implicit network for 100 epochs with optimizer $Adam$, employing a decayed learning rate of $1e-4$ and $3e-3$ decay rate. Each epoch contains $3e6$ samples. Codes and scripts are released in supplementary for reproducibility.

## 5 EXPERIMENTAL RESULTS

Our experiments are conducted on real-world datasets to validate the established layout-object-position association. The data environment is of single-floor residential buildings with multiple rooms which is the common working scenario of household robots widely studied in this field. We employed Matterport3D (Chang et al., 2017) as well as apartment environment (Zhu et al., 2022) dataset to demonstrate that our approach can be generalized in diverse scenarios.

### 5.1 POSITION ATTRIBUTES INFERENCE

To demonstrate the built LOP association integrates positions with layout features, we designed experiments that accept 3D positions as input to infer the region information. For quantitative evaluation, we divided the RGB-D sequences into training and testing sets. The Topo-Field is trained

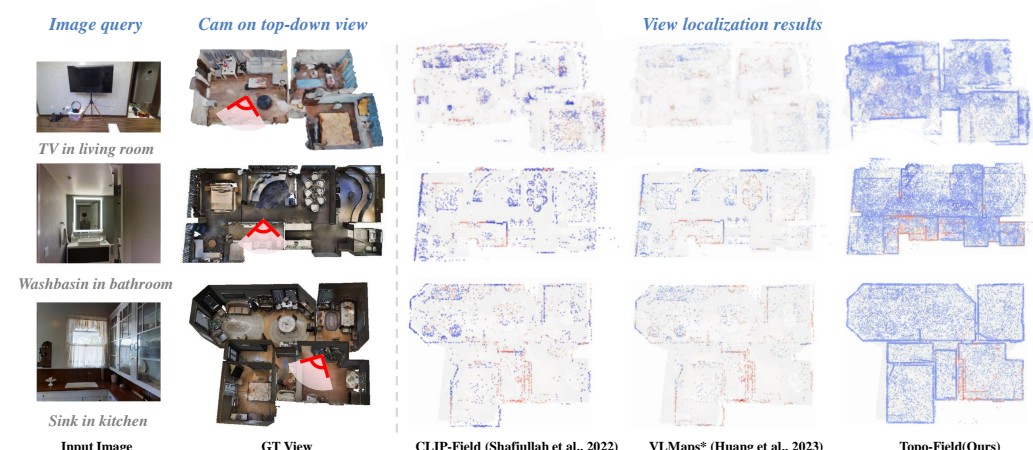

Figure 6: **Qualitative comparison of image query localization** results in heatmaps form among state-of-the-art methods and our method with image input. Our approach localizes the position of queried image in an exact smaller range.

| Methods | Scene1 | Scene2 | Scene3 | Scene4 |
|---|---|---|---|---|
| CLIP-Field | 0.242 | 0.165 | 0.130 | 0.142 |
| Baseline1 | 0.852 | 0.891 | 0.863 | 0.874 |
| Baseline2 | 0.865 | 0.887 | 0.872 | 0.879 |
| Baseline3 | 0.872 | 0.891 | 0.875 | 0.886 |
| Topo-Field | **0.886** | **0.900** | **0.884** | **0.894** |

Table 4: **Ablation of target feature processing pipeline** of the neural field construction. The average region prediction accuracy of sampled points from different scenes on the Matterport3D dataset is used as the metric. Declarations of baselines are clarified in Fig. 7 and Section 4.

according to Section 4.4 on the training set and tested in the test set. As the region inference task can be treated as a multi-class classification task for each input, the accuracy, precision, and F1-score are used as metrics. Tab. A.8 shows the region inference results on 10 real-world scenes in Matterport 3D (Chang et al., 2017) with different scales and layouts indicating the average accuracy exceeds 85%.

## 5.2 Localization with Prompt Queries

**Localization with Text Queries:** For objects of the same category in different regions, we input the textual description of the target object in the form of "object in the region" and infer the specific location of the target, comparing the results with the predictions from current state-of-the-art visual-language algorithms. Fig. 4 demonstrates the advancements of Topo-Field in object localization tasks involving region information, which allows for the localization of specific target objects based on the description and features of the region, while other methods confuse objects from different regions. Tab. 1 shows the quantitative results on 4 scenes of different layouts compared to other methods with an average accuracy of more than 88% and less distance from targets. For the metrics, the average distance ($m$) of predicted point cloud and ground truth point cloud is evaluated, together with counting whether the center of predicted points is in the correct room. Ground truth comes from the Matterport3D provided object instance labels. More results can be seen in the appendix.

**Localization with Image Queries:** To validate the help of region information in the image view localization task. We localize the images from the test set in the trained Topo-Field. Selected views include representative objects of the scene (e.g., TV in the living room) and views with similar-looking objects or context (e.g., bathroom washbasin and kitchen sink) which is challenging. The localization results are shown in Fig. 6 in the form of heatmaps and Tab. 3 shows the quantitative

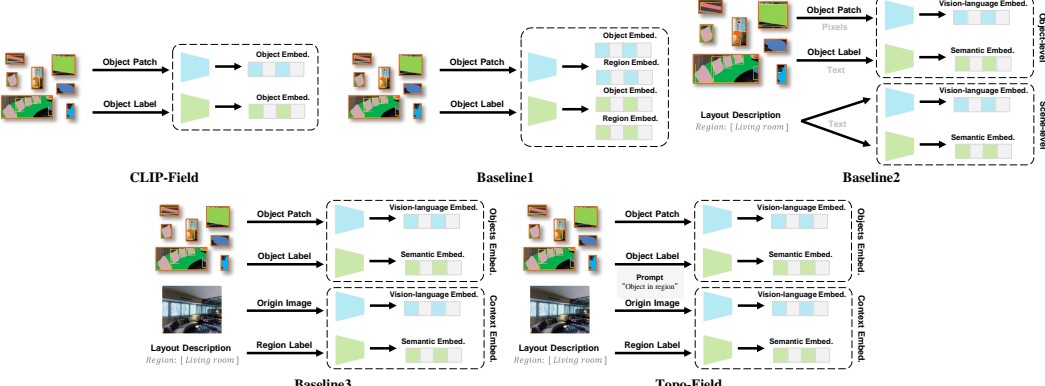

Figure 7: Ablation of our LOP information encoding and feature fusion strategy for target features. (CLIP-Field encoding strategy has been added for comparison.)

results which evaluates the weighted average distance of the target view and localized point cloud among all samples in a scene, using similarity as weight. VLMaps* is a self-implemented version, because origin VLMaps (Huang et al., 2023) does not implement the image localization task. To align with CLIP-Field (Shafiullah et al., 2022) and our work, the LSeg (Li et al., 2022) used in VLMap (Huang et al., 2023) is replaced by CLIP (Radford et al., 2021). The results show that Topo-Field constrains the localization results to a smaller range in the exact region. We sampled more than 40 images on each scene from Apartment (Zhu et al., 2022) and Matterport3D (Chang et al., 2017) dataset. By drawing the predicted camera view on the top-down view, we estimated the localization precision and found that most views can be ranged into a specific view on the target field of view, while other methods struggle to get precise results.

## 5.3 TOPOMETRIC MAP CONSTRUCTION

Fig. 4.3.2 shows an example of the built topometric map. Layout region nodes, object nodes with bounding boxes, and entrance nodes connecting regions are shown with edges representing relationships. A planned navigable path is shown in the graph from an observed view in family room to the TV room sofa in green. The path planning A* algorithm is employed to explore the topological structure to generate waypoints between nodes, and the waypoints are generated with the planning API in Habitat Simulator (Savva et al., 2019) and shown in a line with gradient colors, while different colors indicate different predicted regions of the waypoints.

## 5.4 ABLATION STUDY

Fig. 7 and Tab 4. show the ablation of our neural field LOP encoding strategy and feature fusion where 1) CLIP-Field Shafiullah et al. (2022) means the origin feature encoding strategy that doesn't explicitly consider the layout features. 2) Baseline1 is our first crude approach that directly supervises the learned embedding from the encoded objects with region semantics. 3) Baseline2 encodes the region description to the target vision-language and semantic feature space for supervision. 4) Baseline3 takes the background pixels into account with the region labels. 5) Topo-Field further considers the context of the layout when supervising the object label semantics. These four main versions of our numerous iterations of trying are listed as examples to show our work on the neural field encoding of LOP association.

## 6 CONCLUSION AND LIMITATIONS

Inspired by postrhinal cortex (POR) neurons that prioritize spatial layouts over scene content for cognitive mapping, we propose Topo-Field, which integrates Layout-Object-Position (LOP) associations into a neural field and constructs a topometric map from the learned field for hierarchical robotic scene understanding. However, there are some limitations: 1) While we present a pipeline for topometric map construction, querying and path planning are currently implemented using traditional methods (e.g. A*). Future work will explore using large language models to integrate seman-

tic information for more advanced path planning. 2) Real-world deployment on mobile robots for long-term navigation is needed. 3) Future research will focus on updating and editing the topometric map to accommodate environmental changes.

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

# A APPENDIX

## A.1 SCENE PARTATION EXAMPLE

The scene can be partitioned into different regions using walls as dividers and lines can be aligned to these walls. This is similar in most scenarios, making the annotation of scene regions a straightforward task as shown in Fig. A1.

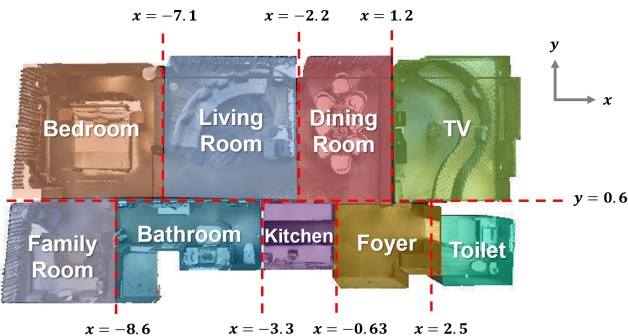

Figure A1: Using walls as dividers to associate lines with them, the scene can be divided into various regions and 3D points can be labeled with related regions easily.

## A.2 VISION-LANGUAGE EMBEDDINGS SIMILARITY OF REGION AND OBJECTS

To demonstrate that the relationship of the vision-language and semantic embeddings for different regions is related to our intuition, we compare the similarity in region-region and object-region form and show the results in Fig. A2. It can be seen that based on general knowledge, cognitively related regions (e.g., the dining room and kitchen) and object-region pairs (e.g., sink and kitchen) are also more correlated in the vision-language and semantic feature spaces.

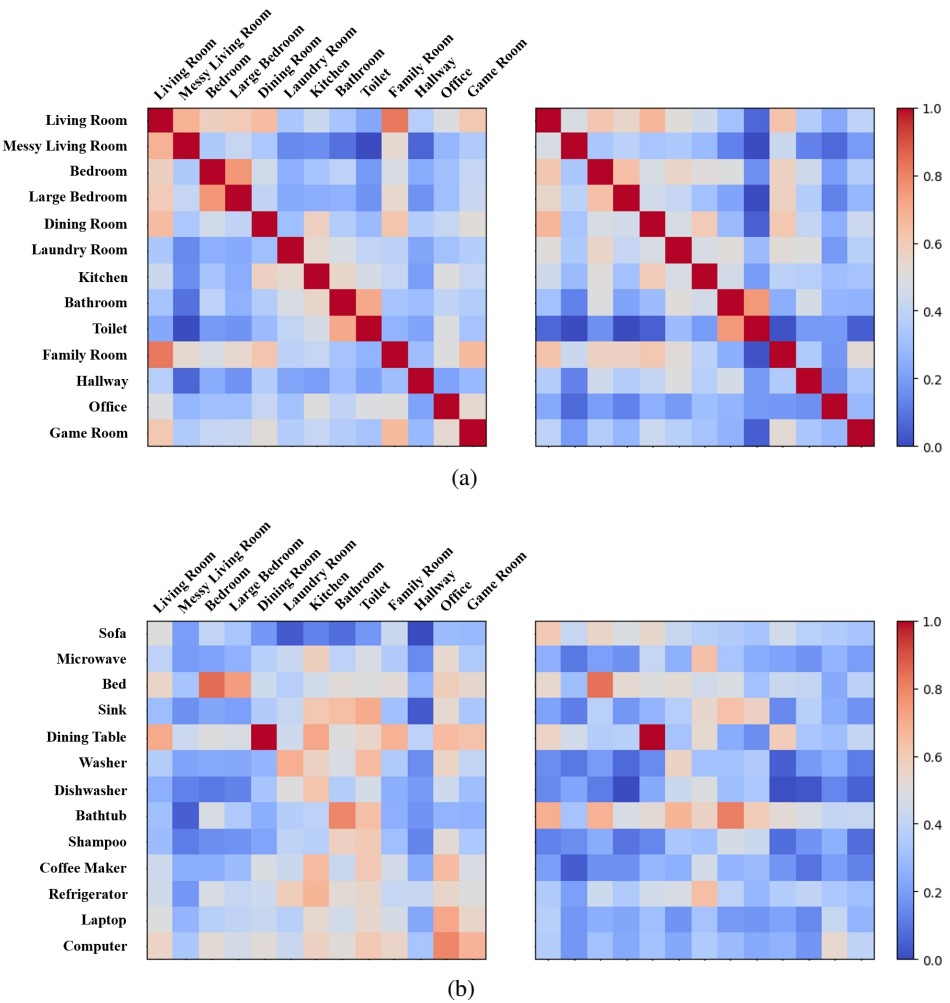

Figure A2: The similarity of a set of region embeddings (as shown in a) and object-region embeddings (as shown in b). The left graph shows the vision-language embedding similarity and the right one shows the semantic embedding similarity.

## A.3 ABLATION STUDY

To explicitly encode the region information, we apply the LVM to process the background pixels out of the object bounding box and LLM to encode the region label text. What's more, for object pixels, object label text is combined with the region text in the form of 'object in the region' before being encoded by LLM. To ablate the contribution of vision-language embeddings from CLIP and semantic embeddings from Sentence-BERT in encoding region features, we compare different weight settings between the v-s embeddings when inferring the regions with 3D position inputs. Results are shown in Fig. A3. It can be seen that both vision-language embeddings and semantic embeddings are indispensable, and weight settings with the greatest results are used for Topo-Field.

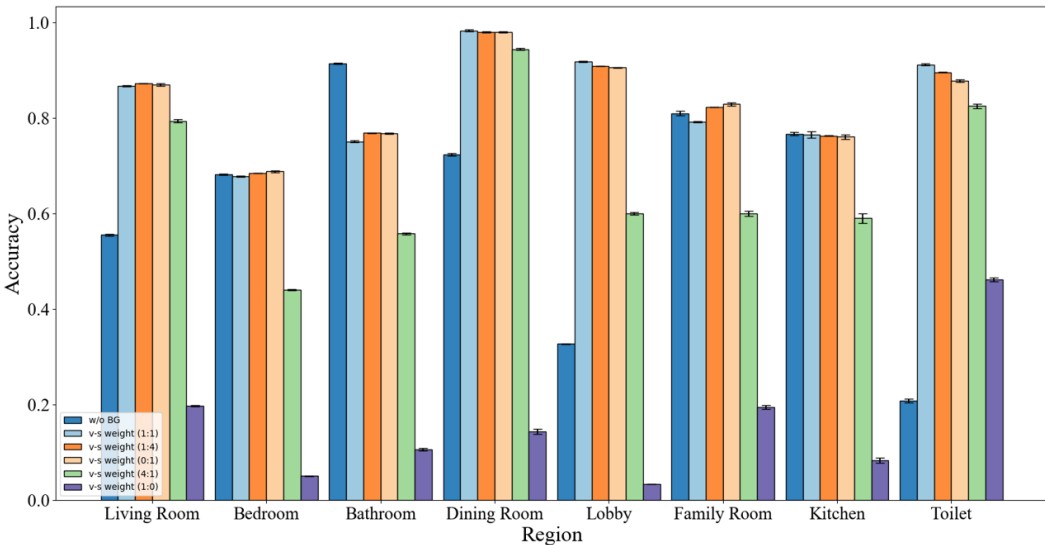

Figure A3: Ablation results on the accuracy of region prediction on Matterport3DChang et al. (2017) with 3D positions input. The w/o BG stands for not encoding background pixels to get region embeddings, and v-s weight ablates the weight of vision-language and semantic embeddings in the embeddings similarity contribution. Error bars show the results among samples from different scenes in Matterport3DChang et al. (2017).

## A.4 HIERARCHICAL APPROACH COMPARISON

Hierarchical scene representation is widely studied with numerous tasks, mainly employing scalable receptive fields and representations to fine-tune results of scalable objects and local relations. As Fig.A.8 shows, VoxFusion introduced octree map with various voxel sizes, LERF employed feature pyramids. As far as we know, few of them explicitly consider the layout level information and the association with objects and positions. This idea comes from recent neuroscience findings, and similar theory has not yet been introduced in scene representations.

## A.5 TOPOMETRIC SEARCH FOR PLANNING

We employ a simple A* approach for planning. Given a topometric graph $G$, the start point $p$, and the target destination object text $t$. First, the belonged region $r$ of $p$ is inferred according to the main paper. The existing objects nodes embeddings are compared with the encoded visual-language and semantic embeddings of $t$ to find the target object node $o$. At the same time, if the region of destination object $r_d$ is declared, the search process would be more simple by directly search among region nodes. Here lists the pseudocode of the employed A*.

## A.6 TOPOMETRIC MAP NODES EXAMPLES

We list the attributes of nodes and edges in the topometric map as example here in Listing $1 - 4$, including the object nodes, region nodes, and edges.

```
1  {
2      "id": 0,
3      "node_type": region,
4      "bbox_extent": [
5          4.163309999999999,
6          4.207343,
7          2.53566175
8      ],
9      "bbox_center": [
```

**Algorithm 1** AStar($G, r, o$)

1: $openSet \leftarrow \{r\}$                 ▷ Set of nodes to be evaluated
2: $cameFrom \leftarrow \{\}$        ▷ Mapping of nodes to their parent nodes
3: $gScore[r] \leftarrow 0$          ▷ Cost from start along best known path
4: $fScore[r] \leftarrow h(r, o)$       ▷ Estimated total cost from start to goal
5: **while** $openSet$ is not empty **do**
6:      $current \leftarrow$ node in $openSet$ with lowest $fScore$ value
7:      **if** $current = o$ **then**
8:          **return** ReconstructPath($cameFrom, o$)
9:      **end if**
10:     remove $current$ from $openSet$
11:     **for** each neighbor $n$ of $current$ **do**
12:        $tentativeGScore \leftarrow gScore[current] + d(current, n)$
13:        **if** $tentativeGScore < gScore[n]$ **then**
14:            $cameFrom[n] \leftarrow current$
15:            $gScore[n] \leftarrow tentativeGScore$
16:            $fScore[n] \leftarrow gScore[n] + h(n, o)$
17:            **if** $n$ not in $openSet$ **then**
18:               add $n$ to $openSet$
19:            **end if**
20:        **end if**
21:     **end for**
22: **end while**
23: **return** "No path found"
24: **function** RECONSTRUCTPATH($cameFrom, current$)
25:      $path \leftarrow [current]$
26:      **while** $current$ is in $cameFrom$ **do**
27:          $current \leftarrow cameFrom[current]$
28:          insert $current$ at the beginning of $path$
29:      **end while**
30:      **return** $path$
31: **end function**

```
10        -8.821845,
11        2.6915385,
12        1.259409125
13    ],
14    "class": "bedroom",
15    "caption": "A bedroom at the northwest of the house with warm
      lighting. Main objects include a bed in the center, a large closet,
      and a dresser at the corner."
16 },
```

Listing 1: Region node

```
1 {
2     "id": 1,
3     "node_type": object,
4     "bbox_extent": [
5         0.3569,
6         0.2297,
7         0.101.8
8     ],
9     "bbox_center": [
10        0.3222,
11        -1.1108,
12        -0.5062
13    ],
14    "class": "picture",
15    "caption": "A white framed picture hanging on the wall."
16 },
```

Listing 2: Object node

```
1 {
2     "id": 0,
3     "node_type": Entrance,
4     "bbox_extent": [
5         0.5,
6         1.6,
7         2.8,
8     ],
9     "bbox_center": [
10        -3.244,
11        -0.276,
12        0.487
13    ],
14    "class": "Entrance",
15    "caption": "Entrance connecting bedroom and living room."
16 },
```

Listing 3: Entrance node

```
1 {
2     "id": 2,
3     "edge_type": region_entrance,
4     "start_node": {
5         "id": 0,
6         "node_type": region,
7         "bbox_extent": [
8             4.163309999999999,
9             4.207343,
10            2.53566175
11        ],
12        "bbox_center": [
13            -8.821845,
14            2.6915385,
15            1.259409125
```

```
16          ],
17          "region_tag": "bedroom"
18      },
19      "end_node": {
20          "id": 0,
21          "node_type": Entrance,
22          "bbox_extent": [
23              0.5,
24              1.6,
25              2.8,
26          ],
27          "bbox_center": [
28              -3.244,
29              -0.276,
30              0.487
31          ],
32          "class": "Entrance",
33          "caption": "Entrance connecting bedroom and living room."
34      },
35      "relationship": connected,
36      "position_relation": "b to the southeast of a",
37      "position_reason": "The x-coordinate of the center of bbox of
        end_node (-3.244) is larger than that of start_node (-8.821845), and
        the y-coordinates of the center of bbox of end_node (-0.276) is less
        than that of start_node (4.207343). Therefore, b is to the southeast
        of a."
38      "caption": "The pathway from bedroom to living room."
39 },
```

Listing 4: Region entrance edge

```
1 {
2      "id": 2,
3      "node_type": object_region,
4      "start_node": {
5          "id": 7,
6          "node_type": object,
7          "bbox_extent": [
8              2.155,
9              2.052,
10             0.883
11         ],
12         "bbox_center": [
13             5.598,
14             2.566,
15             0.136
16         ],
17         "class": "bed",
18         "caption": "a bed with a white comforter and a pillow"
19     },
20     "end_node": {
21         "id": 0,
22         "node_type": region,
23         "bbox_extent": [
24             4.163309999999999,
25             4.207343,
26             2.53566175
27         ],
28         "bbox_center": [
29             -8.821845,
30             2.6915385,
31             1.259409125
32         ],
33         "class": "bedroom"
```

```
34        "caption": "A bedroom at the northwest of the house with warm
      lighting. Main objects include a bed in the center, a large closet,
      and a dresser at the corner."
35      },
36      "relationship": belong,
37      "position_relation": "a in the center of b",
38      "caption": "According to the bbox center position and extent, the bed
       is in the center of bedroom."
39 },
```

Listing 5: Object region edge

### A.7 PROMPT EXAMPLE FOR REGION NODE CONNECTIVITY DESCRIPTION

With topometric mapped nodes, we leverage LLM to describe the connectivity of nodes according to the general knowledge and bounding box 3D position. In listing 5, here we provide a prompt example to describe the connectivity relationship between content objects and regions and set up the edge.

```
1 {
2 DEFAULT_PROMPT_POST = """
3 You are an excellent graph managing agent. Given a graph nodes set of an
      environment,
4 you can explore the relationships of nodes with their attributes and
      build edges among
5 them.
6
7 The input is a list of JSONS describing two types of nodes, including the
       object and
8 region. You need to produce a JSON string (and nothing else) and set up
      edges between them with keys: "relationship", "position_relation" and
       "caption".
9
10 Each of the JSON fields will have the following fields:
11 1. id: a unique number
12 2. node_type: type of this node
13 3. bbox_extent: the 3D bounding box extents
14 4. bbox_center: the 3D bounding box center
15 5. class: an extremely brief description
16 6. caption: a sentence describing node attributes in detail
17
18 Produce a "relationship" field that best describes the relationship of
      the object node and region node. Set "false" if the object is not
      related to the area or is not reasonable, the relationship is refused
      . Produce a
19 "position_relation" field describing the position relationship between
      object and region according to their
20 bounding box information in the 3D space. Before producing the "
      position_relation" field, produce a "caption" field that explains why
       the "position_relation" field is reasonable.
21
22 The built edges should include following fields:
23 1. id: a unique number of each edge in order
24 2. node_type: according to the connected node type in the form "
      start_node\_end_node"
25 3. start_node: keep JSON values of the object node unchanged
26 4. end_node: keep JSON values of the region node unchanged
27 5. relationship
28 6. position_relation
29 7. caption
30 """
```

Listing 6: Prompt example to set up edge with nodes.

## A.8 Additional Experiment Results

Additional experiments results of object localization using text query inputs and view localization using image query inputs. Also, a table is provided showing the metric on exactly each region class from 4 scenes in Matterport3D dataset.

| Regions | Scene1 | | | Scene2 | | | Scene3 | | | Scene4 | | |
|---|---|---|---|---|---|---|---|---|---|---|---|---|
| | Acc. | Pre. | F1 | Acc. | Pre. | F1 | Acc. | Pre. | F1 | Acc. | Pre. | F1 |
| Living Room | 0.948 | 0.970 | 0.959 | 0.870 | 0.881 | 0.875 | 0.778 | 0.810 | 0.793 | 0.902 | 0.949 | 0.925 |
| Bedroom | 0.943 | 0.825 | 0.880 | 0.925 | 0.923 | 0.924 | 0.687 | 0.767 | 0.725 | 0.920 | 0.870 | 0.894 |
| Bathroom | 0.466 | 0.680 | 0.554 | 0.903 | 0.898 | 0.901 | 0.875 | 0.463 | 0.605 | 0.797 | 0.831 | 0.814 |
| Dining Room | - | - | - | 0.961 | 0.794 | 0.870 | 0.774 | 0.732 | 0.752 | 0.933 | 0.887 | 0.910 |
| Lobby | 0.681 | 0.941 | 0.790 | 0.853 | 0.951 | 0.899 | 0.978 | 0.510 | 0.671 | 0.855 | 0.698 | 0.769 |
| Family Room | - | - | - | - | - | - | 0.903 | 0.571 | 0.700 | 0.926 | 0.936 | 0.931 |
| Kitchen | 0.994 | 0.654 | 0.789 | 0.788 | 0.836 | 0.811 | 0.833 | 0.833 | 0.833 | 0.758 | 0.854 | 0.803 |
| Office | - | - | - | 0.969 | 0.848 | 0.905 | - | - | - | 0.953 | 0.883 | 0.917 |
| Toilet | - | - | - | - | - | - | 0.900 | 0.711 | 0.795 | - | - | - |
| Avg. Acc./Samples | 0.886 / 169k | | | 0.900 / 185k | | | 0.884 / 111k | | | 0.894 / 112k | | |

Table 5: Region prediction results on the test set of different scenes from the Matterport3DChang et al. (2017) dataset. Accuracy, precision, and F1 score are used as metrics.

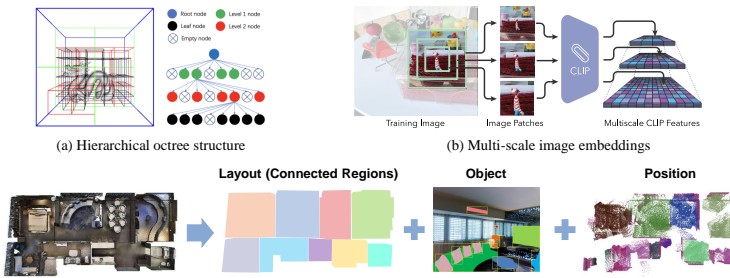

(a) Hierarchical octree structure      (b) Multi-scale image embeddings

(b) The explicitly represented Layout-Object-Position association

Figure A4: The comparison of the hierarchical scene representation strategy against previous works.

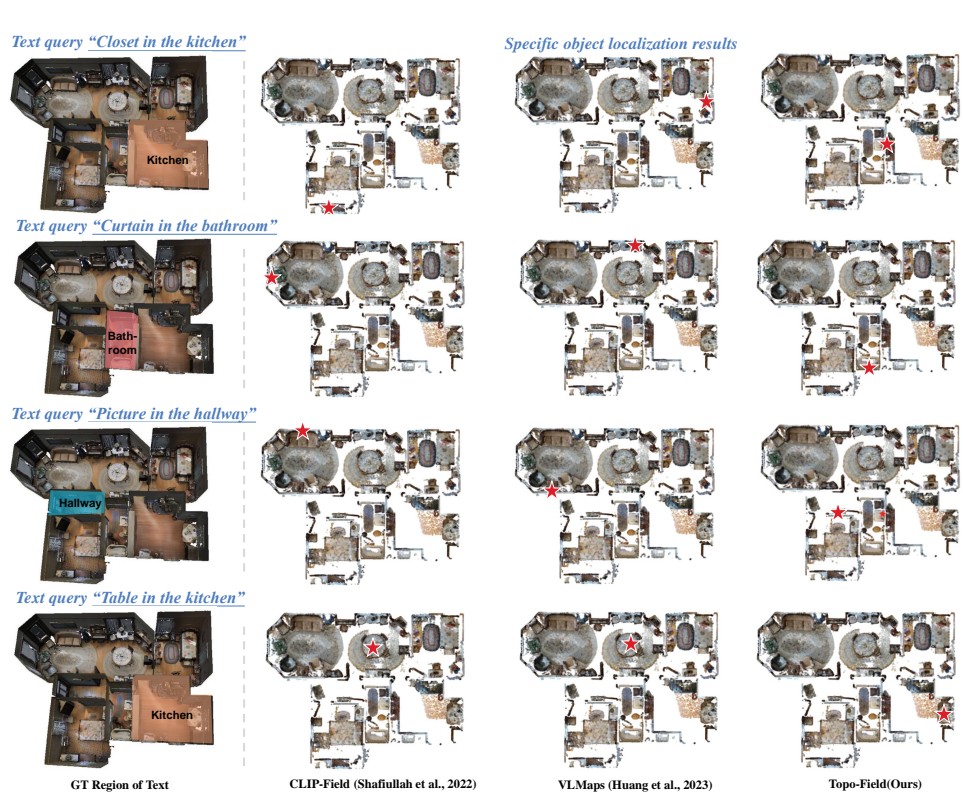

Figure A5: Text query localization on scene 2t7WUuJeko7Chang et al. (2017).

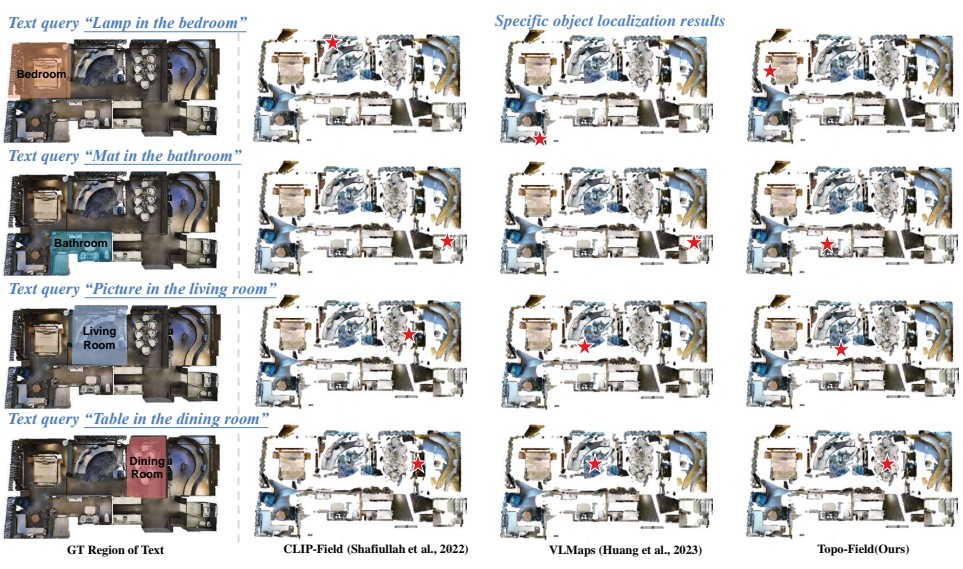

Figure A6: Text query localization on scene 17DRP5sb8fyChang et al. (2017).

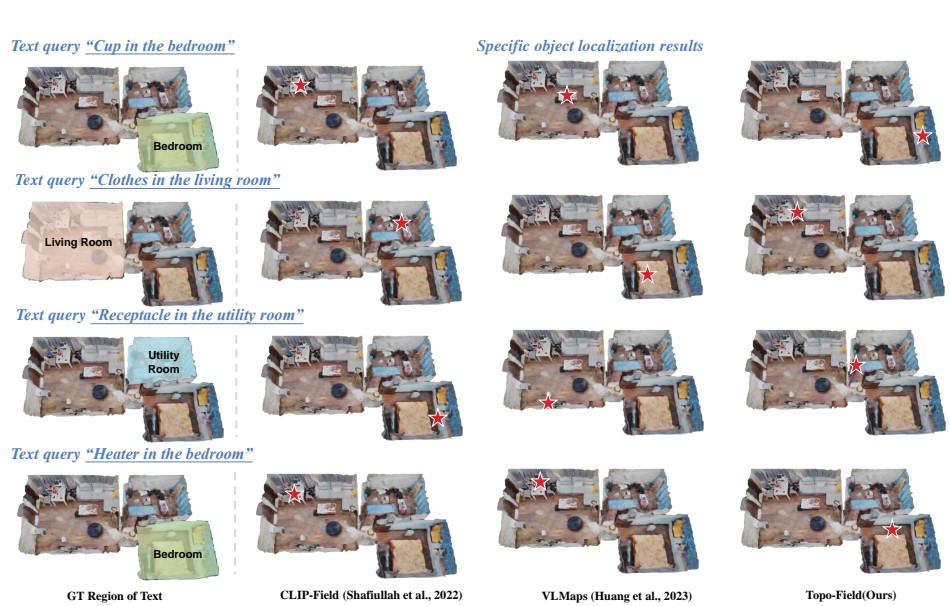

Figure A7: Text query localization on scene ApartmentZhu et al. (2022).

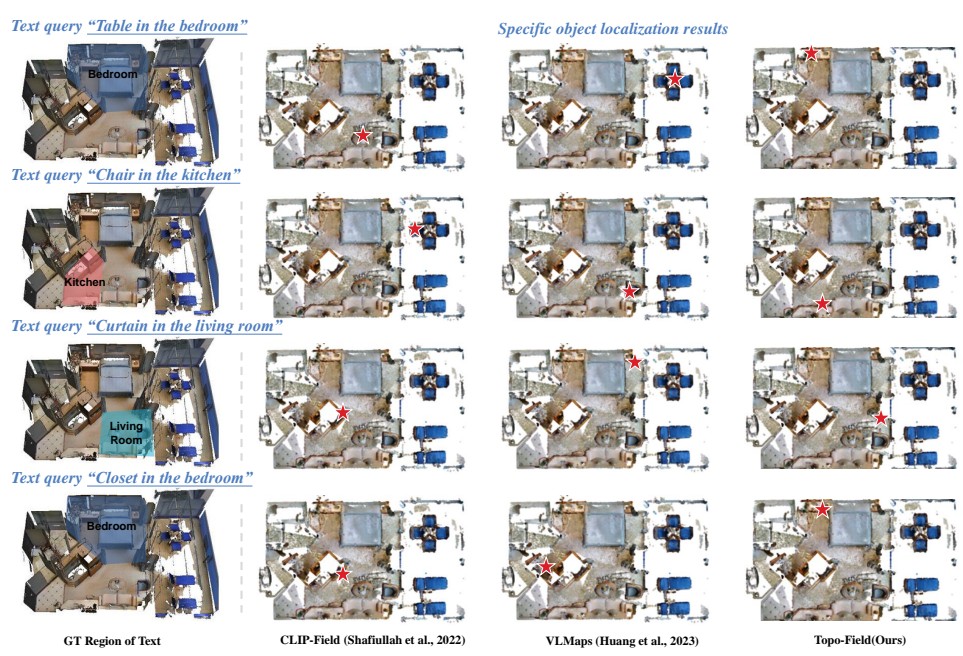

Figure A8: Text query localization on scene HxpKQynjfinChang et al. (2017).

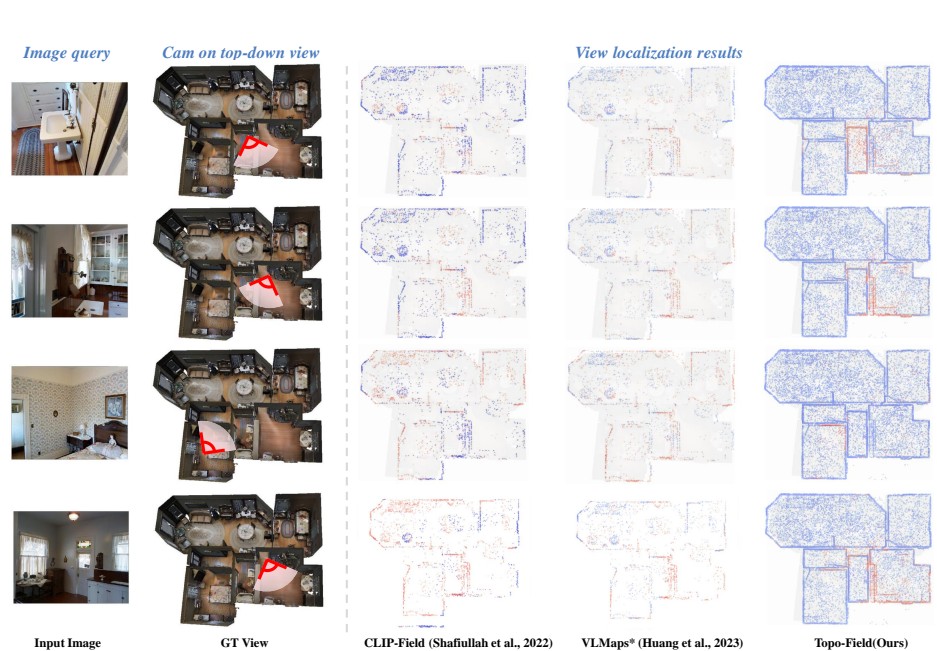

Figure A9: Image query localization on scene 2t7WUuJeko7Chang et al. (2017).

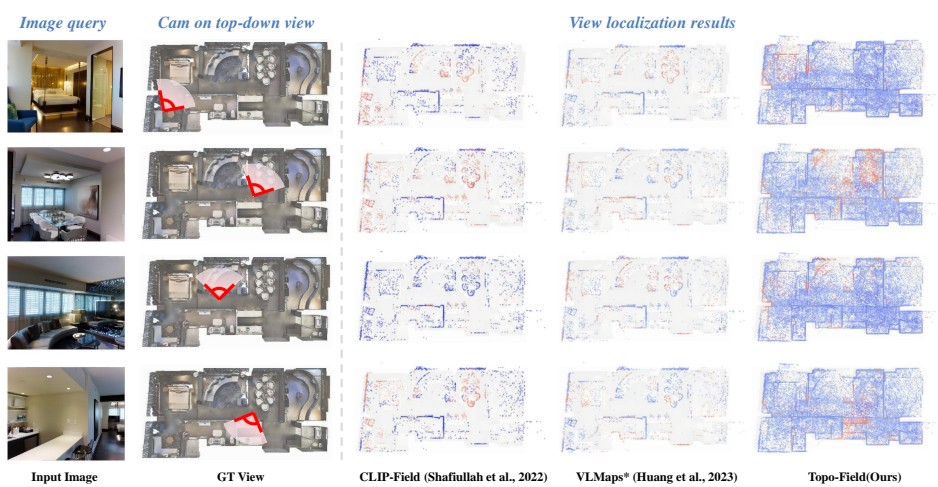

Figure A10: Image query localization on scene 17DRP5sb8fyChang et al. (2017).

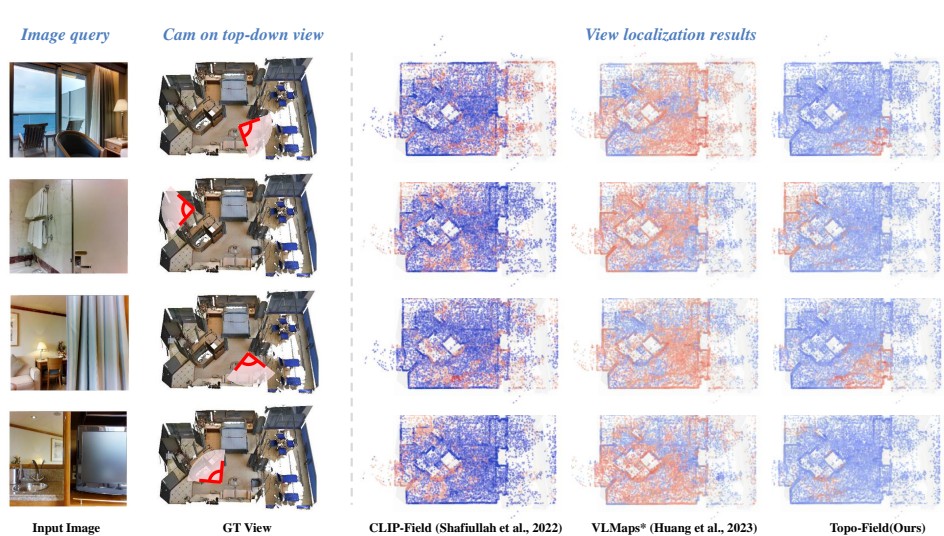

*Image query*    *Cam on top-down view*    *View localization results*

**Input Image**    **GT View**    **CLIP-Field (Shafiullah et al., 2022)**    **VLMaps* (Huang et al., 2023)**    **Topo-Field(Ours)**

Figure A11: Image query localization on scene HxpKQynjfinChang et al. (2017).

