# OpenReview forum: "Topo-Field: Topometric mapping with Brain-inspired Hierarchical Layout-Object-Position Fields"
_ICLR.cc/2025/Conference — ICLR 2025 Conference Withdrawn Submission_

### Official Review · Reviewer_GNDJ · 2024-11-03

**Soundness:** 1
**Presentation:** 1
**Contribution:** 1
**Rating:** 3
**Confidence:** 5

**Summary:**

This paper targets an interesting problem of topometric mapping but is not ready for publishing. The quality is poor regarding writing, organization, annotations, and experimental setups.

**Strengths:**

+ The idea of constructing a topometric map using the implicit neural field is interesting

**Weaknesses:**

The writing is far from satisfactory. The corresponding authors should revise the manuscripts besides the abstract and introduction.
+ Though the paper proposes a Topo-field to integrate layout-object-position, this representation is not clearly presented in Sec. 3. The definition of the topometric map (or the graph structure in Eq. 3) is vague and hard to follow, and the generation of the graph from dense field F (L199) is unclear. Note that the implicit neural field F is similar to previous methods with a distilled feature field, the novelty and the contribution of the proposed method are unclear.
+ The hierarchical structure of point-object-room is common in scene graph generation. However, no relevant work (e.g., CLIO, HOV-SG) is referred to in the related work section or the experiments section.
+ Multiple annotations are not formally defined in the paper (e.g., the functions $C_t, S_t$). The training stage in Sec. 4.4 should be carefully revised to make it clear.
+ The experimental setups lack clear demonstration, and comparisons against recent methods are missing.

**Questions:**

With the issues addressed above, the authors should revise the paper accordingly.

---

> ### Author Response · Authors · 2024-11-19
>
> Thank you for the valuable advice. Sorry for the Ambiguity, proof-reading and formulation clarification on the methodology and settings have been applied and the revised sections are highlighted red in rebuttal pdf.
>
> **For ambiguity problems,** formulations in Section 3, 4 are clarified; Topo-mapping pipeline and matching method are explained in detail in Section 4.3; more training details of neural implicit representation are added in Section 4.4.
>
> **The novelty and the contribution of the proposed method are unclear.**
>
> A cognitive map is a mental representation used by an individual to order personal store of information about spatial environment, and the relationship of its component parts (Tolman, 1948, Psychological review). The cognitive map is embodied by Place cells (O’Keefe et al., 1971, Brain research) and population code in POR is strongly tuned to spatial layout than content (LaChance et al., 2019, Science). Although encoding the layout and contents to form a cognitive map seems a straightforward idea, it has been more than 70 years since the original concept raised.
>
> We mimic the neural mechanisms of spatial representation in three key aspects: 1) The cognitive map corresponds to a topometric map, which uses graph-like representations to encode relationships among its components, e.g. layouts and objects. 2) The population of place cells is analogous to a neural implicit representation with position encoding, enabling location-specific responses. 3) POR, which prioritizes spatial layouts over content, aligns with our spatial layout encoding of connected regions.
>
> We believe this work makes a step forward mimicking and applying mechanisms of spatial cognition on robotics. Our method describes a clear pipeline with details for reproductivity and experiments shows the ability to manage layout-related tasks and the effectiveness of the topo-map.
>
> **No relevant work (e.g., CLIO, HOV-SG) is referred to in the related work section or the experiments section.**
>
> As suggested, the mentioned two very current works (Maggio et al., Oct 2024 RAL; Werby et al., July 2024 RSS) have been added to the related work and discussed. CLIO Maggio et al. (2024) built a task-driven scene graph inspired by Information Bottleneck principle to form task-relevant clusters of primitives. At the same time, HOV-SG Werby et al. (2024) proposed a hierarchical scene understanding pipeline, using feature point cloud clustering of zero-shot embeddings in a fusion scheme and realizing the mapping in an incremental approach. Unlike the incremental mapping and clustering-based graph construction method, we propose to build the topometric map based on querying the trained neural field which serves as knowledge-like memory base, whose nodes and edges include attributes representing object and layout information explicitly learned when training the specific neural encoding.

---

> ### Comment · Reviewer_GNDJ · 2024-11-26
>
> I appreciate the authors' efforts in revising the manuscript and addressing the issues raised. However, it is unfortunate that they continue to emphasize the significance of their 'cognitive realization.' This bio-inspired perspective could be briefly mentioned as the motivation behind the paper, rather than being regarded as a key contribution. The design merely shares a philosophy with the cognitive map, but it can hardly be considered a 'cognitive realization.' The map structures themselves are quite common in the vision community. The information transfer and interactions among the maps have nothing to do with neuroscience. Simply identifying three related concepts separately does not justify calling the proposed method a 'cognitive realization.'
> I strongly recommend that the authors focus on the specific advantages of their proposed method and the particular designs, as the hybrid map with semantic-aware topology is not novel. Take HOV-SG and CLIO for instance:
> * Topometric map: HOV-SG and CLIO maintain Voronoi graphs that explicitly model the topology of free space. These methods also maintain scene graphs to encode objects and their relationships, which represent the realization of a 'topometric map.'
> * Place cells: HOV-SG and CLIO maintain queryable features on the graph nodes, with dense point clouds or triangle meshes for each instance. By constructing a kd-tree of the point cloud/vertice, we can achieve the mapping function
> F (Eq. 1) to retrieve features for any given coordinate x through nearest neighbor search. I don't think a hash-encoded MLP is more 'cognitive' as both can achieve the same function.
> * POR: If the proposed method, with annotated room-type information, can be considered a realization of POR, then HOV-SG and CLIO already maintain explicit nodes for room types within their graph structure (without human annotations).
>
> Therefore, from a high-level perspective, HOV-SG and CLIO effectively achieve what the proposed method claims.

---

> > ### Comment · Reviewer_GNDJ · 2024-11-26
> >
> > Besides the lack of novelty, there is a significant misleading regarding the argument of the learned 'NeRF'. NeRF is an abbreviation for neural **radiance** field, the field should contain color information and is usually optimized through differentiable rendering. The term 'rendered feature' (response to Reviewer ms6U) is also adopted by the authors. However, as illustrated in Fig. 3 and according to Eq. 1, there is no such radiance field (no color and density channels, no differentiable rendering process) at all. The method simply supervises the mapping function F between coordinates and features (L268, 294, 295) given the projected point cloud (coordinate) and the pixel-wise feature pairs.

---

> > > ### Author Response · Authors · 2024-11-26
> > >
> > > We double-checked the revised paper, as mentioned, we employ an implicit neural scene representation in an Instant-NGP (Muller et al., 2022) way rather than a NeRF-based network. The mentioned implicit neural representation or feature field is not the same as neural radiance field.

---

> > ### Author Response · Authors · 2024-11-27
> >
> > Thank you for your effort in making this paper better. Although these two works (HOV-SG and CLIO) are very current, one published in Oct. and one in July, we discussed the differences between our work and theirs in Section 2.2. They construct the map in a feature point cloud clustering and incremental mapping way while we learn a neural implicit representation and construct the map by querying the neural representation. And we are the first to explain the theoretical basis and neuroscience reference to manage the hierarchical encoding of spatial layouts and contents in the form of objects and connected regions.
> >
> > The differences between our approach and their approach are listed as follows to show our novelty: 1) Different map construction approach: CLIO Maggio et al. (2024) built a task-driven scene graph forming task-relevant clusters of primitives. HOV-SG Werby et al. (2024) utilized feature point cloud clustering and managed the mapping in an incremental approach. We learn and represent the spatial embeddings with an implicit neural representation approach and form the topo-map graph by querying the learned representation. 2) Different graph structure: The Voronoi graph is built based on the exploring path-guided point cloud embeddings and clustering process. We query the learned representation in a two-level standard (object or region) separately with fewer vertices. Each vertice clearly represents only one object or region with its attributes. 3) Different scene representation approach: CLIO and HOV-SG form point cloud with features to explicitly represent the scene while we learn an implicit function mapping the 3D positions to the embeddings. That means our approach can interpolate and predict the embeddings of unseen areas and places with sparse or no point cloud.
> >
> > As suggested, we consider to update our contribution as follows:
> >
> > * We develop a brain-inspired Topo-Field, which combines detailed neural scene representation with high-level efficient topometric mapping for hierarchical robotic scene understanding and navigable path planning. Various quantitative and qualitative experiments on real-world datasets are conducted, showing high accuracy and low error in position attributes inference and multi-modal localization tasks. Examples of topometric construction and path planning are also employed.
> >
> > * We explain the theoretical basis and neuroscience reference to manage the hierarchical encoding of spatial layouts and contents in the form of objects and connected regions, according to the spatial mechanism of cognitive map with POR population and place cells.
> >
> > * We propose to learn a Layout-Object-Position associated implicit neural representation with target features from separately encoded object instances and background contexts as objects and layouts. The process is explicitly supervised by LFM-powered strategy with little human labor.
> >
> > * We propose a topometric map construction pipeline by querying the learned neural representation in a two-stage mapping and updating approach, leveraging LLM to validate edges conducted among vertices.

---

> > > ### Comment · Reviewer_GNDJ · 2024-11-27
> > >
> > > I am glad that the authors have presented concrete arguments highlighting their differences against the relevant work. Although both studies appeared on ArXiv before May, I will take HOV-SG as an exemplary case as CLIO was formally accepted after the ICLR submission deadline.
> > >
> > > The authors' response validates my earlier argument that **CONCEPTUALLY**, these scene graph based methods encompass complete forms of topometric maps, place cells, and POR as the proposed method. The primary differences lie in the implementation, and neither the proposed method nor the two relevant works directly replicate the representations found in neuroscience. For further revising the manuscript, the authors are encouraged to provide solid experimental evidence that the proposed method brings advantages given its distinct design choices. The remaining issues are summarized below:
> > > * Theoretical basis. Please specify the paragraph that provides 'the theoretical basis' of the method, as this is regarded as the key contribution of the paper. In the current form, I can only see abstract evidence that similar behaviors exist in the human brain. Is each module necessary? Do different modules communicate in a manner similar to the proposed method? Is such a design superior to other existing designs in the vision community?
> > > * Differences with relevant works. I want to stress the fact that HOV-SG (and CLIO) has the object nodes and region nodes besides the voronoi graph and the point cloud/meshes. The graph structure of the proposed method does not show any difference in terms of the object-region hierarchy but with only missing pieces. The major difference between the proposed methods and the two relevant works is the feature query manner through a hash-encoded network, where the relevant methods maintain the semantic features explicitly on the nodes. The authors are encouraged to provide evidence that query through a network instead of the maintenance on the sparse nodes leads to better accuracy or efficiency.
> > > * Interpolate embeddings on unseen areas: Please provide concrete evidence that such interpolation/prediction behavior leads to practical advantages instead of providing noisy and inaccurate semantics.
> > > * Map construction. Please clarify your map construction process. I note the following terms and definitions: 'the learned neural field' F (eq.1, F is formally denoted as the topo-field later at L375), a 'topometric map' G = (V, E) (eq. 3), a 'Topo-Field' with $g$ and $h$ (L269). As the authors claim that 'we construct the topometric map in a mapping and updating strategy based on the learned Topo-Field F', I wish the authors could clarify how the topometric map is constructed in the MAPPING phase based on the Topo-Field F as I don't find the involvement of F in the paragraph between L377 and L414.
> > > * Navigable path. Please explain how the maintained map representation facilitates 'navigable path planning' (claimed in the first contribution). Note that the topo-field only contains the mapping between the coordinate and the semantic features, and the topometric map only contains sparse nodes and edges without a metric map (as defined in Eq. 4 & 5). How is A* algorithm (L568) applied to this map representation to generate the 'navigable path'?
> > >
> > > There are also numerous issues regarding the experiments.
> > > * The authors first present the region inference results in Sec. 5.1. I don't understand why this evaluation is conducted as the region is manually annotated.
> > > * Regarding the localization with text queries in Sec. 5.2, the failure cases of other methods (as shown in Fig. 4) consistently demonstrate that the text queries identify correct object semantics but in incorrect rooms. The results do not convincingly demonstrate the superiority of the proposed method, and the comparison is unfair since the room type information requires manual annotation in the paper.
> > > * The paper mentions 10 test scenes (Table 2) but provides comparative results for only four of them (Tables 3 and 4). Please clarify this inconsistency.
> > > * Please specify the scene IDs of Scene 1-10 so that future work can make fair comparisons.

---

> ### Author Response · Authors · 2024-11-28
>
> The **BRAIN-INSPIRED** Topo-field draws intuitive lessons from biological evidence, with the primary differences lying in our perspective and then leading to different subsequent implementations. We were inspired by evidence that neurons in the postrhinal cortex (POR) exhibit a preference for the spatial layout of local scenes, defined by the geometry of regions, such as a room's boundaries. Based on this, we abstract the spatial representation of regions to align with our spatial layout encoding of connected regions. However, our goal is not to elucidate the neural mechanisms of POR firing patterns but rather to leverage these principles for practical spatial representation.
>
> HOV-SG (and CLIO) and Topo-field share similar intuitive ideas about graph-like representations with nodes for rooms and objects, likely stemming from shared human common sense. Supported by the preference for spatial representations observed in POR, we specifically conceptualize rooms as spatial layouts of local scenes, establishing a one-to-one node correspondence with regions in our topometric map. In contrast, the Voronoi graph in HOV-SG serves primarily to provide traversable areas with more detailed nodes and edges, whereas our topometric map emphasizes integrating spatial layout information and semantic details about rooms and objects. While the underlying ideas may seem similar, the distinct starting points result in significantly different implementation methods.
>
> For the remaining issues:
>
> * We incorporate the spatial layout information into Topo-Field supported by the biological evidence that the POR population prefers the spatial layout of local scene, corresponding to the geometry of regions, connected rooms in the topometric map.
>
> * Neurons in POR prefer layouts of local scenes inspiring the one-to-one correspondence of rooms where the map maintains fewer nodes with explicit room and object semantic information.
>
> * Obviously, our proposed method could take any 3D position in the map as input to predict the semantic information, unlike the point cloud or meshes.
>
> * We build the topometric map by sampling among positions to get the semantic and metric information of room and object nodes as shown in Figure 2(c).
>
> * As shown in Eq. 4 & 5 the bounding_box includes the center and extent information of objects and regions clearly providing the metric for the topometric map, which can be used for A* path planning.
>
> For the issues in the experiments:
>
> * We learn and evaluate the annotated region information to validate that the neural representation is able to conduct a relationship between observed image background contexts and region vision-language embeddings. It is realized by mapping back-projected 3D locations to region embeddings.
>
> * It is our contribution to explicitly represent and learn the region layouts. For other compared methods, it does not mean that they haven't considered the context information, the open-world embeddings
> distilled in their feature fields implicitly include the objects and contexts. The experiments show our advantages in explicitly
> constructing the layouts and relationships with objects and 3D positions.
>
> * We choose 2 large scale and 2 small scale scenes as representative scenes in Matterport3D dataset for query
> localization as mentioned in Section 5.1.
>
> * It would be included in the paper and our provided reproducible code as demo if accepted.

---

### Official Review · Reviewer_wzd7 · 2024-11-03

**Soundness:** 2
**Presentation:** 3
**Contribution:** 2
**Rating:** 5
**Confidence:** 3

**Summary:**

This paper presents a method for training a neural implicit field that utilizes supervisory signals from pre-trained foundation models to capture semantic features. The proposed model is applicable to several critical downstream tasks in robotics, including text/image query localization, semantic navigation, and path planning. Experimental results demonstrate significant improvements in performance metrics, supported by qualitative evidence.

**Strengths:**

The paper addresses a compelling problem in semantic mapping and its applications for enabling robots to navigate real-world environments. The experimental results demonstrate impressive improvements in performance. Additionally, the supplementary materials, such as the code snippets and prompts, enhance the understanding of the proposed method's details and implementation.

**Weaknesses:**

Despite its potential, the system heavily relies on various input types, such as annotated room maps and camera poses, as well as off-the-shelf object detection methods for generating bounding boxes and masks. This dependence poses challenges in real-world applications, where inaccuracies in these inputs can lead to errors. Additionally, the system's reliance on ChatGPT complicates debugging and explanation when errors occur in complex real-world environments.

Encoding semantic information and supervising it with pre-trained features alleviates some annotation burdens; however, this approach is already a common practice in the field of implicit representation for semantic mapping [1][2]. The overall system resembles a large engineering project, making it challenging to distill its theoretical contributions.

[1] V. Tschernezki, I. Laina, D. Larlus and A. Vedaldi, "Neural Feature Fusion Fields: 3D Distillation of Self-Supervised 2D Image Representations," 2022 International Conference on 3D Vision (3DV), Prague, Czech Republic, 2022, pp. 443-453, doi: 10.1109/3DV57658.2022.00056.
[2] Zhu, Siting, et al. "Sni-slam: Semantic neural implicit slam." Proceedings of the IEEE/CVF Conference on Computer Vision and Pattern Recognition. 2024.

Table 4 could benefit from clearer labeling, where Baselines 1-4 are not explicitly defined. A reference to Figure 7 could help.

The authors frequently reference the postrhinal cortex from the biological literature, but the connection to the proposed method is not clearly articulated. Topological mapping is indeed a common computer vision task relevant to navigation.

**Questions:**

The experimental results are impressive when compared to baseline performances; however, it is unclear whether the benchmarks used are new proposed by the authors or following existing ones, which raises concerns about the fairness of the evaluation. What are the primary factors driving the significant improvements?

Computation: the paper mentions a large batch size of 12,544. It would be helpful to clarify what specific data is contained within this batch size.

---

> ### Author Response · Authors · 2024-11-19
>
> Thank you for agreeing on our contribution, after proofreading and formulation clarification, the paper includes more details and is helpful for understanding and reproducing. Revised version is attached in rebuttal with revised sections highlighted in red.
>
> **For ambiguity problem,** Table 4 and Figure 7 are revised, referenced, and described in section 5.4.
>
> For other problems:
>
> **The system heavily relies on various input types, poses challenges in real-world applications. The system's reliance on ChatGPT complicates debugging and explanation when errors occur in complex real-world environments.**
>
> As using NeRF, the posed images are needed and mentioned in Section 4.1, COLMAP (Sch¨onberger & Frahm, 2016) is widely employed method to provide these problems. This is the same with most other reconstruction and scene representation works. And off-the-shelf large foundation models are widely employed by approaches to provide labels without human labor. Indeed, besides Matterport3D which is a real-world dataset, a real-world apartment environment (Zhu et al., 2022) is also employed for evaluation, which proves effectiveness and practicability.
> For usage of GPT to help evaluate the vertices relationship, additional details have been clarified in Section 4.3.2. As mentioned, the GPT is used in our approach to filter out unreasonable relationships and check vertices relations which mainly decide problems like (1) whether a bike located in a bedroom is possible (2) the 3D location relationship of b-box[x1,y1,z1] and b-box[x2,y2,z2]. The GPT is well-prompted as in appendix and this would not cause big error.
>
> **The overall system resembles a large engineering project, making it challenging to distill its theoretical contributions, the connection to the proposed method is not clearly articulated:**
>
> A cognitive map is a mental representation used by an individual to order personal store of information about spatial environment, and the relationship of its component parts (Tolman, 1948, Psychological review). The cognitive map is embodied by Place cells (O’Keefe et al., 1971, Brain research) and population code in POR is strongly tuned to spatial layout than content (LaChance et al., 2019, Science). Although encoding the layout and contents to form a cognitive map seems a straightforward idea, it has been more than 70 years since the original concept raised.
>
> We mimic the neural mechanisms of spatial representation in three key aspects: 1) The cognitive map corresponds to a topometric map, which uses graph-like representations to encode relationships among its components, e.g. layouts and objects. 2) The population of place cells is analogous to a neural implicit representation with position encoding, enabling location-specific responses. 3) POR, which prioritizes spatial layouts over content, aligns with our spatial layout encoding of connected regions.
>
> We believe this work makes a step forward mimicking and applying mechanisms of spatial cognition on robotics. Our method describes a clear pipeline with details for reproductivity and experiments shows the ability to manage layout-related tasks and the effectiveness of the topo-map.
>
> **Whether the benchmarks used are new proposed by the authors or following existing ones, which raises concerns about the fairness of the evaluation.**
>
> Existing scene representation methods either evaluate semantic segmentation results like mIOU (Shafiullah et al., 2022; Huang et al., 2023) or simply evaluate the localization accuracy (Kerr et al., 2023). Based on the existing evaluation strategy and for detailed quantification, we improve this metric by employing point cloud distance of prediction and target and region localization accuracy according to our proposal. For fairness, our method and compared ones share the same input, metrics, and evaluation strategy.
>
> **Computation: the paper mentions a large batch size of 12,544. It would be helpful to clarify what specific data is contained within this batch size.**
>
> As mentioned in (Shafiullah et al., 2022), a larger batch size helps the CLIP (Radford et al., 2021) series works reduce the variance in the contrastive loss function. As a reliable baseline, CLIP-Field (Shafiullah et al., 2022) used a batch size of 12,544 to maximize the VRAM usage. For fairness, we keep align with the same settings in our approach for the settings of MHE network.

---

> > ### Comment · Reviewer_wzd7 · 2024-11-25
> > **Discussion**
> >
> > Thank you to the authors for responding to the questions. However, I feel that some of the issues have not been fully addressed. Please see the details below.
> >
> > 1. ChatGPT
> >
> > Thank you for including the prompts in the Appendix. However, I am still unclear about the exact inputs and demo outputs for the system.
> >
> > Additionally, the phrase "would not cause big error" is somewhat ambiguous. Could you clarify what potential errors might arise, and how the system can address these to ensure robust deployment in real-world, open-world environments?
> >
> > 2. Cognitive Map
> >
> > While I understand that the theory of cognitive maps is well-established, I am unclear about what novel aspects are introduced into this paper from a cognitive perspective, as opposed to existing well-defined knowledge in the topology domain. While the narrative is compelling, I feel there is a weak logical connection between the ideas.
> >
> > 3. Contributions and Related Work
> >
> > Most importantly, the authors have not addressed my primary concern regarding the contribution of this work relative to other papers. Specifically:
> >
> > "Encoding semantic information and supervising it with pre-trained features alleviates some annotation burdens; however, this approach is already a common practice in the field of implicit representation for semantic mapping [1][2]."
> >
> > 3. Evaluation
> >
> > I understand the evaluation metrics presented. However, my key concern lies with the dataset splits. Specifically, I would like to know if the training and evaluation data used are consistent with those commonly employed in previously published work in this area.

---

> ### Author Response · Authors · 2024-11-26
>
> Thank you for the active discussion, for remaining problems:
>
> **ChatGPT:** As declared in Section 4.3.2, we leverage LLM to help validate the
> construction of topometric graph edges. The input is two JSON files including
> region vertices and object vertices whose attributes have been declared in Section 4.3.2.
> and samples are given in Appendix A.6. With the input and prompts listed, LLM is supposed
> to output a new JSON file including the graph edges among vertices, where edge attributes
> are declared in Section 4.3.2 and samples are given in Appendix A.6.
>
> As for the phrase "would not cause big error", we mean in our practice we have not found
> obvious error. One shortcoming may come as follows, the position relation of regions is defined
> as one of: 1) a is to the east of b, 2) a is to the west of b, 3) a is to the north of b, 4) a is to
> the south of b. In fact, a room could be located to the southeast-south of the other room, LLM
> may decide the relationship to be 1) a is to the east of b. As mentioned "As mentioned, the GPT
> is used in our approach to filter out unreasonable relationships and check vertices relations
> which mainly decide problems like (1) whether a bike located in a bedroom is possible (2) the
> 3D location relationship of b-box[x1,y1,z1] and b-box[x2,y2,z2]." LLM has the common knowledge
> and ability to deal with these easy problems.
>
> **Cognitive Map:** As discussed in Section 2.2, traditional topo-map didn't include semantics (Zhang, 2015; Zhang et al., 2015; Garrote et al., 2018; Oleynikova et al., 2018; Badino et al., 2012). Concept-graph (Gu et al., 2024) makes a step forward utilizing LFM to model the object structure with a topo map, which introduces open-world semantics. CLIO Maggio et al. (2024) and HOV-SG Werby et al. (2024) propose using feature point cloud clustering and mapping in an incremental approach, which is not a cognitive inspired approach.
>
> On the contrary, based on the mental representation of cognitive map, LaChance et al. in 2019 discoveried that population code in POR is strongly tuned to spatial layout than content. More recently, Zeng et al. in 2022 proposed that geometry representations of local layouts relative to environmental centers are needed to form a high-level cognitive map from egocentric perception to allocentric understanding. We propose to encode spatial layout and contents with a layout-object-position field. By querying the neural representation from egocentric perceptions, we form an allocentric high-level graph-like topometric map representing layouts with connected regions relative to its centers.
>
> **Contributions and Related Work:** In a word,
> most semantic feature fields learned in existing methods (Zhi et al., 2021; Fan et al., 2022; Xie et
> al., 2021; Shafiullah et al., 2022; Huang et al., 2023; Kerr et al., 2023) focus on object
> semantics but do not include layout-level features. Works like RegionPLC (Yang et al., 2023)  considered region
> information by fusing multi-model features but no explicit representation of layout features is learned.
> The discussion has been included in Section 2.1 and 2.2 with more detail.
>
> **Evaluation:** The dataset splits used for our localization evaluation follows a general setting which is about 4:1 to 5:1 like most learning-based localization works (Kendall et al., 2016).

---

### Official Review · Reviewer_dJX7 · 2024-11-05

**Soundness:** 2
**Presentation:** 2
**Contribution:** 2
**Rating:** 5
**Confidence:** 3

**Summary:**

This article proposes a novel approach for encoding scene information into a topometric map, for improving localisation and planning. The proposed approach is based on a Layout-Object-Position (LOP) approach. Layout information from knowledge of the environment's rooms. Object information from semantic segmentation (Detic) and a joint encoding of the segmented object patch using clip and of the object-region labels using Sentence-BERT. Finally, position information is produced by a 3D reconstruction of the scene using Multi-scale Hashing Encoding (MHE). This information is combined into a single Topometric map coined Topo-Field.
The proposed method is evaluated for the inference of position attributes and localisation and appears to clearly outperform the presented baselines.

**Strengths:**

- The combination of structural and semantic information in a way that is efficient for robotic system to query and plan on is a critical problem for robotics.
- The approach seems to perform very well on the evaluation, clearly outperforming the presented baselines on those tasks.
- The proposed approach is also reasonable in computational terms, as all experiments were performed on a single GPU (no information is given on training time though).

**Weaknesses:**

- The description of the approach is lacking specifics, and the reader has to infer the architecture and information flow from the provided diagrams rather than formal description in mathematical and algorithmic terms.
- It is not fully clear how the partitioning of the environment (location) into rooms is performed and how well it would generalise to new environments.
- The motivation of the work from neuroscience is interesting, but remains very vague. Little discussion is provided on how well the proposed approach may model the neural structures it claims to be inspired by.
- The performance is very good compared to the discussed baselines, but it would seem that the proposed appraoch also benefits from significantly more task-specific information for those tasks (ie, the room information is provided directly). This is not a critical issue in my view, but it would be good to discuss the limitations of the presented baselines and issue of fairness of comparison some more.
- I note that the reference for Reimers & Gurewych should probably cite the published version of the article rather than the pre-print.

**Questions:**

- In line 235: what is C and S? I assume it is the output of CLIP and Sentence-BERT? How are the regions r_p defined?
- In line 239: what is m in this equation?
- In page 5, line 241: It would seem that the partitioning of the space requires human labeling? If that is the case, it is a significant limitation of the approach.
- In line 242: Could you clarify the sentence "the predicted implicit representation outputs are targeted to match the features from the pre-trained models separately", what it means in practice or how this is achieved. I assume this is what is described in 4.2, but it would be good to make it unambiguous if that is the case, as F is not referenced in that section.
- Could you make the description in Section 4.2 more specific and formal? The only description of inputs/outputs and process we are provided is via the diagram in figure 1, it would be good to have a proper formal description of the process, description of the architecture, and format/dimensionality of inputs and outputs for each component, as well as a formal algorithm
- In line 254: Could you provide a more in-depth argument for using MHE? The computational cost of standard NeRFs is well known, but is MHE the only possible solution? How does it compare with other fast approaches discussed in the literature, like, for example, Gaussian Splatting?
- In line 258: Could you describe the mapping in more formal terms? Fig. 2 only provides a schematic description of the process.
- In line 268: How is the similarity between E_pi and {C_R, S_R} calculated? It would be good to have a formal equation for this operation.

---

> ### Author Response · Authors · 2024-11-19
>
> Thank you for agreeing on our contribution and carefully reading to make this paper better. Sorry for the Ambiguity, proof-reading and formulation clarification on the methodology have been applied and the revised sections is highlighted red in rebuttal pdf.
>
> **For ambiguity problems**, environment partitioning detail is described in Section 4.1; discussion of proposed approach with bio-inspired theory is added in Introduction and contribution; citation error is fixed; formulations in Section 4.1, 4.2, 4.3 4.4 are checked and clarified; neural encoding including MHE and separate heads are declared in Section 4.2; Topo-mapping pipeline and matching method is explained in detail in Section 4.3.
>
> As for other questions:
>
> **How well the proposed approach may model the neural structures it claims to be inspired by:**
>
> A cognitive map is a mental representation used by an individual to order personal store of information about spatial environment, and the relationship of its component parts (Tolman, 1948, Psychological review). The cognitive map is embodied by Place cells (O’Keefe et al., 1971, Brain research) and population code in POR is strongly tuned to spatial layout than content (LaChance et al., 2019, Science). Although encoding the layout and contents to form a cognitive map seems a straightforward idea, it has been more than 70 years since the original concept raised.
>
> We mimic the neural mechanisms of spatial representation in three key aspects: 1) The cognitive map corresponds to a topometric map, which uses graph-like representations to encode relationships among its components, e.g. layouts and objects. 2) The population of place cells is analogous to a neural implicit representation with position encoding, enabling location-specific responses. 3) POR, which prioritizes spatial layouts over content, aligns with our spatial layout encoding of connected regions.
>
> We believe this work makes a step forward mimicking and applying mechanisms of spatial cognition on robotics. Our method describes a clear pipeline with details for reproductivity and experiments shows the ability to manage layout-related tasks and the effectiveness of the topo-map.
>
> **It would seem that the proposed approach also benefits from significantly more task-specific information for those tasks:**
>
> For fairness, the same input, metrics, and evaluation strategy are employed for our method and all compared ones. So the better performance comes from our method explicitly modeling the layout structural and objective information and the hierarchical integration, powered by the whole pipeline mentioned before.
>
> **It would seem that the partitioning of the space requires human labeling? If that is the case, it is a significant limitation of the approach.**
>
> We agree that labeling the scene regions needs human labor. However, in fact, partitioning the buildings needs little human labor, where in most human-made buildings spatial layouts are easily available divided by straight walls. Clarified in section 4.1. Layout information is available in datasets like Matterport3D. However, if not provided, the region distribution can be easily annotated taking little human labor. As in our practice, region annotation of a house with 8 rooms only takes 3 min by drawing lines from top-down view according to walls to form a rule to separate (x,y) coordinates, bounding 3D points to different regions.
>
> **Is MHE the only possible solution? How does it compare with other fast approaches discussed in the literature, like, for example, Gaussian Splatting:**
>
> Surely the MHE is not the only possible solution as you mention, other methods like plenoxels, octree, feature grids or others could be solution. Gaussian Splatting is a hot explicit scene representation research recently, however we believe the NeRF, being an implicit way to encode information, is more likely a neural way mimicking the bio-encoding way. There’s no research proving NeRF represented implicit strategy or GS represented explicit way which is better. Further comparison could be a good research, however would not be included in this paper.

---

> > ### Comment · Reviewer_dJX7 · 2024-11-27
> >
> > Thanks for your response to my earlier questions.
> >
> > I don't think the answer really addresses my question concerning additional information in the authors' method: The other baselines do not have access to the layout information and therefore are at an explicit disadvantage. This seems to be confirmed by the fact that all of the model variations in the ablation study seem to outperform the other baselines with a significant margin (Table 4). It would be useful to devise an additional evaluation protocol that would allow to disentangle the performance increase due to the simplification of the problem by the added layout information and the one due to the proposed architecture.
> >
> > The motivation of the model from the evidence of places cells in POR is just too vague in my opinion. The paper would need some evaluation demonstrating that the proposed model and encoding are similar to the population coding observed in POR.
> > Similarly, the statement that NeRF is more biologically plausible as an implicit coding would require a more rigorous argumentation based on the known properties from neuroscience evidence.

---

> ### Author Response · Authors · 2024-11-28
>
> Thank you for your insightful comments and suggestions.
>
> Regarding the motivation of our model, we were inspired by evidence that neurons in the postrhinal cortex (POR) exhibit a preference for the spatial layout of local scenes, which is determined by the geometry of regions, such as a room's boundaries. Based on this neurobiological evidence, we abstracted the spatial representation of regions to align with our spatial layout encoding of connected regions. This encoding aims to capture the spatial structure in a way that is consistent with the principles observed in POR.
>
> For the integration of layout information in our method, the addition to the Topo-field is motivated by this brain-inspired approach, incorporating the spatial layouts connected by regions. To evaluate the impact of this addition, we conducted comparisons with and without the layout information, as shown in our ablation studies. These results demonstrate the contribution of layout information to the model's performance. As for architecture improvement, ablations from Baseline1 to Topo-Field already include as shown in Figure 7 and Table 4.

---

### Official Review · Reviewer_ms6U · 2024-11-06

**Soundness:** 2
**Presentation:** 1
**Contribution:** 2
**Rating:** 5
**Confidence:** 4

**Summary:**

The paper introduces Topo-Field, a framework designed to enhance mobile robot navigation by integrating detailed semantic information about layouts, objects, and their positions (LOP) into a neural field representation. Interestly, such structure is inspired by the role of postrhinal cortex neurons on spatial layout. By querying a learned NeRF, Topo-Field constructs a semantically rich yet computationally efficient topometric map for hierarchical robotic scene understanding. Experimental results demonstrate its effectiveness in tasks like position inference, localization, and planning, bridging the gap between detailed scene understanding and efficient robotic navigation.

**Strengths:**

1. The authors tackle the problem of hierarchical robotic scene understanding, which is an interesting and important topic
2. The proposed LOP is bio-inspired, to me this concept seems interesting.

**Weaknesses:**

**Unclear descriptions of target feature processing in Sec 4.1**
1. How do you know if a 3D point belongs to the object, or the background? Do you use the GT annotations from the dataset? (the Matterport3D you show has that information I believe?)
2. For the background features, you will get only a single feature for each image. How do you fuse those features from different views?
3. Also, isn’t it making more sense to take per-pixel CLIP features from models like LSeg/OpenSeg, and fuse that information?

**Unclear descriptions of neural scene encoding in Sec 4.2**
1. Related to the questions above. In this section you mention that there are object-level local features and layout-level region features and MHE seems to be a good representation for learning such hierarchy. However, how exactly do you learn these two set of features respectively under MHE? No details there
2. To learn MHE or NeRF in general, you need to actually shoot a ray for each pixel and sample along the ray. The final features are the weighted sum of all values along the ray, with volume rendering. How do you make sure your features on the 3D surface point are exactly the feature you render?

**Unclear Topometric Mapping in Sec 4.3**
1. Line 309, what is ${C_t, S_t}$? What are the differences to ${C_R, S_R}$ (I know this is the embeddings for region) in Line 304, and ${C_I, S_I}$ in Line 314? You did not specify them before. It is confusing and making it hard to understand
2. Figure 3 (b) does not really match with what you write in “localization with text/image query” between Line 306-318. In the figure, all you get are the per-point features, and try to match with query features, omitting many important details in your description.
3. “Matching” in Figure 3 is never really discussed. What kind of matching? Do you mean calculating the cosine similarity among the features, and take the one with the highest score?

**Text query localization in experiments**
1. How do you decide the similarity threshold for the bounding box? Do you need to choose a different threshold for each text query? My own experience is that, it is not really possible to get a single threshold for every query.
2. One more thing: once you have the right threshold, how exactly do you get the bounding boxes out from thresholding?
3. What are the “samples” in Table 1?
4. How many queries are you considering for each scene, and how do you obtain the GT? Same question applies to Table 3 as well.

**Image query localization in experiments**
if I understand correctly, you show the heatmap of the query. You claim that “Topo-Field constrains the localization results to a smaller range in the exact region”. However, that is not really true to me. If you look at the washbasin in the bathroom, you also have many points highlighted in other regions, like kitchen, and even some points in the bedroom. In such a case, how can you get such good numbers in Table 3?

**Ablation study**
How come your ablation in Table 4 is only evaluating the region prediction accuracy, which does not even require most parts of your methods (objects, the graph you build, etc). Why not evaluate on other things as well? And even that, your default strategy seems not outperform much over any of baselines, even the very simple baseline 1 in some scenes.

**Writing**
- Overall I think the writing is not good since many things are not justified well.
- There are so many cases when the author writs (), a space is not added before, e.g. L214 …mapping(SLAM), L259 Multi-layer Perceptron(MLP), etc.

**Questions:**

It would be very important if you can justify those points in the weakness part.

---

> ### Author Response · Authors · 2024-11-19
>
> Thank you for showing interest in our novel idea and carefully reading to make this paper better. Sorry for the Ambiguity, we carefully apply proof-reading and formulation clarification on the methodology and the revised sections is highlighted red in rebuttal pdf.
>
> **For ambiguity problems**, the pixel-wise encoding strategy is clarified in Section 4.1; neural encoding including MHE and separate heads are declared in Section 4.2; Topo-mapping pipeline and matching method is explained in detail in Section 4.3; formulation in the figure 2, 3 has been clarified; space is added before all ().
>
> For other questions:
>
> **Bounding-box in text query localization:** Actually, it’s not the general bounding-box from object detection, we filter the points with similarity over threshold (0.6 in our practice), and simply draw a bounding box to cover these points for visualization. ‘Samples’ is the number of text queries, which is clarified in revised pdf. Ground truth comes from the object instance labels from Matterport3D. For table 3, we’ve mentioned in Section 5.2 that more than 40 images are sampled from each scene, ground truth comes from back-projecting image pixels into 3D according to ground truth pose and depth.
>
> **Image query localization:** Table 3 shows weighted average distance among all samples in a scene, using similarity as weight. Given that few points in orange may appear in other rooms as noise, the max distance of a single point from these points (similarity would be 0.3\~0.6) would be less than 6\~8 m, which counts relative little.
>
> **Ablation study:** For better understanding, we add the origin feature encoding strategy in CLIP-Field (Shafiullah et al., 2022) into Figure 7 and Table 4 as comparison to show the improvement more obviously. As for the improvement from Baseline1 to current Topo-Field, this metric is evaluated on more than 100~200k position samples on each scene, even if the number seems not growing in a large scale (1% ~ 3%), it’s a robust and obvious progress.
>
> For the metrics, while there have been many works distilling object features and comparing semantic segmentation, our work focuses on the layout-level encoding and the integration with object-level. As you can see, the single object feature encoding branch nearly remains the same in our work. As for topometric graph, we aim to provide a pipeline to build this map based on neural implicit representation and evaluate its effectiveness with tradition graph-based plan method. More quantification for optimization and evaluation on robotic would be our under-going future work.
>
> By the way, for fairness, the same input, metrics, and evaluation strategy are employed for our method and compared ones.

---

> > ### Comment · Reviewer_ms6U · 2024-11-25
> >
> > I appreciate the authors' effort for all the responses! Still, there are many points that I am not convinced or not clear.
> >
> > For example, there are still some unclear points from your method, I am just listing a few below.
> > 1. Line 243, why CLIP can give you “per-pixel” feature? Isn’t that a global feature for each image? You still did not really answer my questions of using “LSeg/OpenSeg” for per-pixel feature.
> > 2. You still did not justify my question “How do you make sure your features on the 3D surface point are exactly the feature you render?” In Sec 4.2, I still don’t understand it
> >
> > Moreover, there are still unjustified points in experiments:
> > 1. Ok, so you choose a cosine similarity score threshold of 0.6, why this value? How can such a single threshold value work well? For example, my personal experience with the CLIP cosine similarity score is that, threshold=0.4 might work well for “bed”, but might not be working well with “bed with a pillow on it” (just an example). Therefore, an experiment to justify your choice of threshold, and an explanation of why a single value of 0.6 can work well for all scenarios is necessary
> > 2. Now I understand better why your image localization performance is good, but in your paper, you still did not add an explanation
> > 3. Ablation: you partially answer my question, I appreciate it. However, you still did not answer the key question: why your ablation is only on the region prediction, not any other parts of your method (objects, the graphs, etc)
> >
> > Based on my concerns above, I am afraid that the paper right now has not reached my standard for publication. Please incorporate all the comments from every reviewer and submit them to the next venue.

---

> > > ### Author Response · Authors · 2024-11-26
> > >
> > > Thank you for the active discussion, for remaining problems:
> > >
> > > **Feature encoding:** I now understand your concern. CLIP provides per-pixel feature and pixels in the bounding-box share the same feature of the object. As for background, all pixels outside the bounding-box share the same feature to represent the unified concept of "region". On the contrary, LSeg tends to provide per-pixel feature and each pixel feature varies. However, encoding the region information of an image is more likely an image classification task rather than a segmentation task, which aims to supervise all pixels in the background with a unified region label. It's hard to convergent scattered features from different pixels to a unified concept.
> > >
> > > **Feature contrastive learning:** For “How do you make sure your features on the 3D surface point are exactly the feature you render?”, we further add a more detailed discussion to train the neural representation in Section 4.2. In the way discussed in 4.2 about neural encoding and 4.1 about target feature processing, given a posed RGB-D image, the target feature of each pixel is processed as mentioned in 4.1 denoted as $\mathcal{E}\{(e_v, e_s)\}$ (features on the 3D surface point). At the same time, the related pixel in depth image is back-projected into 3D space according to depth and pose value, denoted as $p$, and processed by the process mentioned in 4.2 to form $F(p)=\{f_v, f_s\}$ (rendered feature). A contrastive loss is conducted between $\{(e_v, e_s)\}$ and $\{f_v, f_s\}$ to train the neural representation. Training details are declared in Section 4.4. Figure 2 clearly shows the feature contrastive learning pipeline.
> > >
> > > **Threshold:** As mentioned, "we filter the points with similarity over threshold (0.6 in our practice), and simply draw a bounding box to cover these points for visualization". Generally, there are more than 30\~50 points on a single object, so there's a tolerance on the threshold choice. As for the exact value 0.6, it is an empirical choice based on our experiments on Matterport3D dataset among tens of scenes we've tested. In our practice values among 0.4\~0.6 does not bring too much disturbance on results.
> > >
> > > **Image query localization revise in paper:** The metric and sample strategy declaration has been added in Section 5.2.
> > >
> > > **Ablation:** Our main contribution is proposing the allocentric layout-based scene encoding with the neural representation approach and constructing a topological map based on this. As can be seen in Fig. 7 the object encoding branch keeps nearly the same with previous method, so we do not ablate the object related metrics like semantic segmentation which would be nearly the same with previous works. As for graph, we propose a topometric map construction pipeline based on the learned neural representation based on the queried object and region features. Consequently, the topo-map construction result relies on the learned object and region embedding metrics. Evaluating and improving the topo-map are our undergoing future work on graph-based path planning and locomotion which would not be included in this paper.

---

### Note · Authors · 2024-12-11

I have read and agree with the venue's withdrawal policy on behalf of myself and my co-authors.